# MORC2 mediates transcriptional regulation through liquid-liquid phase separation

Yanshen Zhang[1,2,3†], Weiya Xu[1,2†], Wenxiu Duan[1,2†], Yu Wei[1,2,3], Wenli Jiang[1,2,3], Feng Zhu[1], Chengdong Huang[1,2*], Chao Wang[1,2,3*], Yihui Bi[4*]

[1]Hefei National Research Center for Physical Sciences at the Microscale, Center for Advanced Interdisciplinary Science and Biomedicine of IHM, Division of Life Sciences and Medicine, University of Science and Technology of China, Hefei, China; [2]Ministry of Education Key Laboratory for Membraneless Organelles and Cellular Dynamics, Biomedical Sciences and Health Laboratory of Anhui Province, Division of Life Sciences and Medicine, University of Science and Technology of China, Hefei, China; [3]Department of Neurology, the First Affiliated Hospital of USTC, Division of Life Sciences and Medicine, University of Science and Technology of China, Hefei, China; [4]Department of Orthopedics, The Second Affiliated Hospital of Anhui Medical University, Hefei, China

*For correspondence:
huangcd@ustc.edu.cn (CH);
cwangust@ustc.edu.cn (CW);
yihuibi@ahmu.edu.cn (YB)

†These authors contributed equally to this work

## eLife Assessment

This **important** study has demonstrated that MORC2 undergoes phase separation in cells and established multiple interactions responsible for the phase separation. Upon revision, the data generally provide **solid** support to the claim that MORC2 condensates are functionally relevant in gene regulation and begins to demonstrate the importance of the physical properties of biological condensates. Nevertheless, there remains some weakness in the connection between condensates and function.

**Abstract** MORC2 is a chromatin-associated ATPase essential for transcriptional silencing and genome stability, yet the biophysical principles governing its regulatory activity remain elusive. Here, we demonstrate that full-length MORC2 undergoes biomolecular condensation to form dynamic nuclear assemblies, a process fundamentally required for its repressor function. Endogenous MORC2 forms discrete, dynamic condensates in neurons from Morc2a[EGFP] chimeric mice, supporting the physiological relevance of these assemblies in vivo. Mechanistically, a 3.1 Å crystal structure of coiled-coil 3 (CC3) identifies a dimeric scaffold that serves as a structural hub, while multivalent 'sticker' interactions between an intrinsically disordered region (IDR) and a newly defined IDR-binding domain (IBD) drive condensation. We show that DNA acts as a molecular scaffold that triggers MORC2 condensation, which in turn allosterically stimulates its ATPase activity. Critically, by employing a 'killswitch' strategy to decouple assembly from internal fluidity, we reveal that only dynamic MORC2 condensates, not static aggregates or condensation-deficient mutants, can restore transcriptional regulation in MORC2-knockout cells. Furthermore, pathogenic variants linked to CMT2Z and SMA differentially perturb these material properties and enzymatic turnover, providing a mechanistic link between condensate dysregulation and human neuropathies. Together, our findings establish a DNA-templated condensation mechanism for MORC2 and provide a molecular framework for understanding how the material state of chromatin-associated machinery dictates gene regulation and disease pathogenesis.

## Introduction

Chromatin dynamically regulates DNA accessibility to orchestrate essential processes such as transcription, replication, and DNA repair (*Alabert and Groth, 2012*; *Ishihara et al., 2021*; *Downs and Gasser, 2024*; *Blackledge and Klose, 2021*; *Li et al., 2012*). These processes are dynamically regulated by chromatin-associated complexes, such as the human silencing hub (HUSH) complex (*Müller and Helin, 2024*), which control the accessibility of genetic material to regulatory machinery. The Microrchidia (MORC) protein family, characterized by a conserved N-terminal GHL-ATPase domain and a structurally diverse C-terminal region, regulates gene expression and chromatin organization (*Zhong et al., 2023*; *Xue et al., 2021*; *Weiser et al., 2017*; *Moissiard et al., 2012*). These C-terminal domains are thought to contribute to dimerization and functional specificity (*Wang et al., 2021*; *Tchasovnikarova et al., 2017*; *Douse et al., 2018*; *Xie et al., 2019*). Notably, MORC2 has been identified as a component of the HUSH complex and is required for HUSH-mediated transcriptional repression (*Tchasovnikarova et al., 2017*).

GHL ATPases, including DNA gyrase, Hsp90, and MutL, rely on ATP-dependent dimerization to drive conformational changes essential for substrate engagement and enzymatic regulation (*Morais Cabral et al., 1997*; *Chen et al., 2020*; *Liu et al., 2016*; *Ortega et al., 2021*; *Lee et al., 2021*). Similar mechanisms have been proposed for MORC family members. In humans, the MORC family comprises five members-MORC1, MORC2, MORC3, MORC4, and SMCHD1, and different MORC paralogs target distinct genomic loci, including transposable elements, viral DNA, and long-exon genes (*Zhong et al., 2023*; *Pastor et al., 2014*; *Dopkins et al., 2022*). The conserved N-terminal GHL-ATPase domain of MORC family proteins has been extensively characterized, whereas the evolutionarily diversified, multi-domain C-terminal regions are thought to act as key determinants of chromatin targeting and functional specificity. Despite these shared structural features, the molecular basis for MORC-mediated transcriptional regulation remains poorly understood, partly due to limited structural insight into the full-length proteins and C-terminal regions.

MORC2 has recently been identified as a key regulator of chromatin architecture and gene silencing, primarily through its recruitment to HUSH-target loci via C-terminal CC2 domain-mediated interactions with TASOR and MPP8 (*Tchasovnikarova et al., 2017*). While its silencing activity was initially attributed to the N-terminal ATPase domain, genetic complementation studies have revealed that the CW domain and all three coiled-coil (CC) regions are also indispensable for its function (*Tchasovnikarova et al., 2017*; *Douse et al., 2018*). Beyond its structured domains, sequence analyses and structural predictions have uncovered an extended intrinsically disordered region (IDR) spanning the central to C-terminal regions of the protein, suggesting potential regulatory mechanisms by facilitating multivalent and dynamic interactions beyond enzymatic activity (*Fendler et al., 2025*).

Transcriptional silencing is frequently spatially organized within membrane-less nuclear compartments, often driven by the process of biomolecular condensation (*Leidescher et al., 2022*; *Liang and Cai, 2023*). Sequence analyses of MORC2 reveal an extended IDR spanning the central to C-terminal domains, suggesting a capacity for the multivalent, weak interactions that underpin phase separation (*Boeynaems et al., 2018*; *Yoshizawa et al., 2018*). While IDRs are common in chromatin-associated proteins, their presence alone does not guarantee condensation; rather, phase separation often emerges from the coordinated interplay between disordered segments and structured oligomerization hubs. How such mesoscale organization modulates MORC2's enzymatic turnover and its ability to remodel the genome remains a fundamental question in the field.

In this study, we demonstrate that MORC2 undergoes phase separation to form dynamic nuclear condensates. We define the structural basis for this behavior, identifying a dimeric CC3 scaffold that, together with multivalent interactions between the IDR and a newly identified IDR-binding domain (IBD), drives condensation. We further show that DNA binding acts as a molecular scaffold to promote MORC2 assembly, which in turn allosterically enhances its ATPase activity. Using a combination of cellular 'killswitch' assays and in vivo models, we establish that the dynamic material properties of MORC2 condensates, rather than mere protein recruitment, are strictly required for transcriptional regulation. Finally, we provide mechanistic insights into how neuropathic disease-associated mutations in the MORC2 N-terminus perturb this biophysical balance, linking condensate dysregulation to human pathology.

## Results

### Coiled-coil 3 (CC3) mediates dimerization of full-length MORC2

To determine whether MORC2 has N- and C-terminal dimerization interfaces similar to those in other GHL-type ATPases (*Verba et al., 2016*), which previously identified the role of CC3 (previously designated as CC4) in MORC2 dimerization (*Xie et al., 2019*), we established an in vitro system to purify full-length (FL) human MORC2 (MORC2FL, residues 1–1032) from HEK293F cells, yielding nucleic acid-free, highly pure proteins for structural and functional analyses (*Figure 1a and b*). Analytical size-exclusion chromatography coupled with static light scattering (SLS) demonstrated that MORC2FL forms dimers, with a molecular weight of approximately 239±1 kDa under-buffer conditions containing 1000 mM NaCl even in the absence of ATP (*Figure 1c*). Notably, the addition of 601 DNA resulted in a detectable increase in molecular weight (303±2 kDa), indicative of DNA-protein complex formation, but did not disrupt the dimeric assembly of MORC2 in the same buffer condition (*Figure 1c*).

To elucidate the atomic structure of MORC2FL, we initially employed negative-stain electron microscopy to reveal that MORC2FL particles were poorly resolved and exhibited considerable heterogeneity in buffer containing 300 mM NaCl. Notably, a subset of particles displayed a morphology consistent with dimer formation (*Figure 1—figure supplement 1a*). To further investigate this, we performed structural prediction of MORC2FL using AlphaFold3 and identified a predicted structure that closely resembled the observed reconstruction (*Figure 1—figure supplement 1b*). We then examined the oligomerization properties of various MORC2 constructs. MORC2$^{1-900}$ (FL∆CC3-IBD domain) was found to exist as a monomer, whereas MORC2$^{901-1032}$ and MORC2$^{901-1003}$ exhibited dimeric properties, confirming the critical role of the CC3 domain (residues 901–1003) in mediating MORC2 dimerization (*Figure 1c*, *Figure 1—figure supplement 1c*).

To gain further structural insights, we determined the 3.1 Å crystal structure of the CC3 dimer using X-ray diffraction (*Supplementary file 1*). The structure revealed that the dimer interface is stabilized predominantly by hydrophobic interactions. Three key hydrophobic residues, L911, L915, and F922, form distinct layers within the dimer interface (layers 1–3, *Figure 1e*, and *Figure 1—figure supplement 1e, f*). The side chains of these residues are buried within the dimerization interface, creating a hydrophobic core critical for maintaining the stability of the dimer.

We systematically introduced single-point mutations in these hydrophobic residues to assess their contributions to dimerization. SLS analysis demonstrated that the wild-type (WT) MORC2$^{901-1032}$ (with Trx-tag) fragment predominantly exists as a dimer (~61 kDa). In contrast, all three key mutants (L911Q, L915Q, and F922Q) displayed altered molecular weight distributions indicative of disrupted dimer formation and enhanced higher-order oligomerization (more than 61 kDa; *Figure 1—figure supplement 1g*). However, none of these mutations completely disrupted dimerization or converted the dimer to a monomeric state (*Figure 1—figure supplement 1g*). Further analyses of additional single-site mutants (e.g. I908Q, Y921A, F951Q, Y954A, L958Q) showed minimal effects on dimer stability, underscoring the central importance of L915 and its neighboring hydrophobic layers in stabilizing the CC3 dimer (*Figure 1—figure supplement 1g*). Moreover, we examined the dimeric status of truncated CC3 constructs (residues 901–975), which confirmed that the CC3 N-terminal region is sufficient to drive dimer formation (*Figure 1f*). To delineate the structural basis of these interactions, we mapped the positions of hydrophobic residues within the CC3 backbone (*Figure 1—figure supplement 1e*). The distances between interacting residues, such as I908-I908' (3.4 Å), L911-L915' (3.7 Å), F922-F922' (3.9 Å), and L958-L958' (4.0 Å), revealed a tightly packed hydrophobic network. The robustness of this network ensures the integrity of the CC3 dimer, even under varying salt concentration and protein concentration tested in vitro conditions. These findings collectively suggest that the CC3 domain plays a critical role in MORC2 dimerization. The dimeric assembly of CC3 is essential for maintaining the structural integrity of the protein, which likely underpins its functional activities, including associations with chromatin (*Xie et al., 2019*).

### MORC2 forms condensates both in vivo and in vitro

In the predicted structure of MORC2FL, we observed an extended unstructured region at the C-terminus of the protein (*Figure 1—figure supplement 1b*). This region is classified as an intrinsically disordered region (IDR), a protein architecture frequently reported to facilitate multivalent interactions and can act as a scaffold for phase separation (*Boeynaems et al., 2018*; *Alberti et al., 2019*).

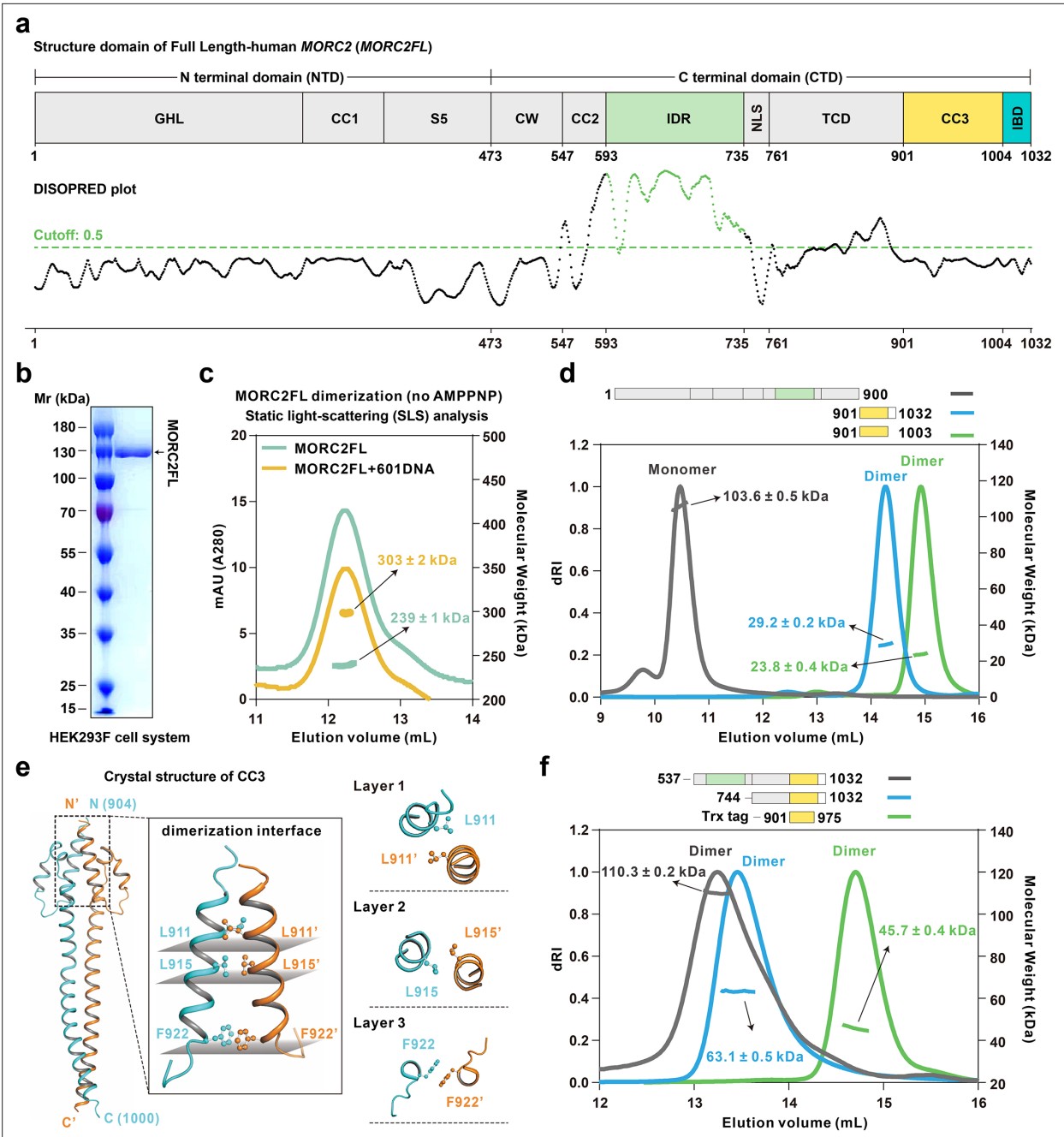

**Figure 1.** The CC3 domain serves as the structural basis for full-length MORC2 dimerization. (**a**) The domain organization of human MORC2FL, highlighting the ATPase module (GHL, CC1 and S5 domains, collectively termed NTD), the C-terminal domains (CTD) with coiled-coil regions (CC2 and CC3), CW-type zinc finger (CW), intrinsically disordered region (IDR, marked by a fresh green line by IUPRED2 prediction), nuclear localization signal (NLS), Tudor-chromodomain (TCD), and a 30-residues tail domain (IBD). (**b**) Purification and characterization of full-length human MORC2 in HEK293F cell system. SDS-PAGE analysis of the purified protein demonstrates its purity and distribution. (**c**) Static light-scattering (SLS) analysis shows that MORC2 forms dimers in the absence of ATP, with a molecular weight of 239±1 kDa in standard buffer (mint green line) and 303±2 kDa in the presence of 601 DNA (sunflower yellow line). (**d**) SLS analysis reveals the oligomerization states of MORC2 truncations. Full-length MORC2 lacking CC3-IBD (1-900) is monomeric, while CC3-IBD (901–1032) and CC3 alone (901–1003) form stable dimers. (**e**) Crystal structural of the CC3 dimer. Hydrophobic residues contributing to dimer stability (L911, L915, and F922) are highlighted using ball-and-stick representations. Top view of the hydrophobic core of the CC3 dimer. Layers of hydrophobic contacts stabilize the dimer interface, illustrated with paired residues (e.g. Layer 1: L911-L911', Layer 2: L915-L915', and Layer 3: F922-F922'). (**f**) Additional SLS analysis confirms that fragments spanning residues 537–1032, 744–1032, and Trx-901–975 (CC3 truncation) adopt dimeric conformations.

The online version of this article includes the following source data and figure supplement(s) for figure 1:

*Figure 1 continued on next page*

*Figure 1 continued*

**Source data 1.** Original files for SDS-PAGE gel analysis displayed in *Figure 1b*.

**Source data 2.** PDF file containing original SDS-PAGE for *Figure 1b*, indicating the relevant bands and treatments.

**Figure supplement 1.** Full-length MORC2 is a dimeric protein, with dimerization mediated by its C-terminal CC3 domain.

Specifically, IUPred2A analysis predicts that the residues 593–735 predominantly exhibit high disorder scores, prompting us to define this segment as the IDR domain (*Figure 1a*).

In vitro, sedimentation assays showed that when the salt concentration was reduced to 150 mM, more than half of the EGFP fused MORC2FL (EGFP-MORC2FL) precipitated into the pellet fraction (*Figure 2a and b*). Fluorescence imaging revealed that at this near-physiological ionic strength, EGFP-MORC2FL spontaneously demixed to form well-defined spherical droplets (*Figure 2c and d*). These droplets exhibited a characteristic polydispersity in size and showed significant coarsening as protein concentration increased (*Figure 2—figure supplement 1a*). Sedimentation assays further showed that the pellet fraction of EGFP-MORC2FL after centrifugation was concentration-dependent (*Figure 2—figure supplement 1b*). The EGFP-MORC2FL-enriched condensates exhibited dynamic behavior, including droplet fusion (*Figure 2—figure supplement 1c*) and fluorescence recovery after photobleaching (FRAP; *Figure 2e and f*), indicating a dynamic state.

Using antibody labeling, we observed that endogenous MORC2 forms discrete, dense punctate structures within the nuclei of HeLa cells (*Figure 2g*). The specificity of these puncta was confirmed by their absence in *MORC2*-knockout (KO) HeLa cells (*Figure 2—figure supplement 1d, e*), suggesting that MORC2 may natively organize into higher-order nuclear condensates under physiological conditions. To further characterize these assemblies, we transfected HeLa cells with EGFP-MORC2FL and observed the formation of prominent, droplet-like nuclear condensates (*Figure 2h*). Notably, consistent with the endogenous staining patterns, EGFP-MORC2FL condensates preferentially localized to regions of reduced chromatin density, as visualized by the H2A-mCherry signal. This distinct spatial distribution implies a potential role for MORC2 condensates in organizing or regulating the heterochromatin landscape. FRAP analysis of these EGFP-MORC2FL puncta revealed fluorescence recovery on a timescale of seconds (*Figure 2i and j*). Furthermore, these puncta underwent active fusion events (*Figure 2—figure supplement 1f*), a hallmark of dynamic liquid-like condensates with a highly mobile internal environment.

To determine whether MORC2 forms biomolecular condensates under physiological conditions, we generated mice expressing EGFP-MORC2 protein from the endogenous locus (*Figure 2—figure supplement 2a*). We focus on neurons in the brain and spinal cord as missense mutations in *MORC2* cause neuropathies including Charcot-Marie-Tooth and spinal muscular atrophy disease (*Wang et al., 2021*; *Douse et al., 2018*). Imaging of NeuN-positive neurons revealed that EGFP-MORC2FL condensates occur in neurons under endogenous expression level (*Figure 2k*). FRAP analysis in freshly prepared brain slices further demonstrated rapid and partial fluorescence recovery, supporting the dynamic nature of endogenous MORC2 assemblies (*Figure 2l*). In addition, unlike MORC3, which shows cell cycle-dependent nuclear puncta formation (*Zhang et al., 2019*), MORC2 condensates were absent in mitotic cells (*Figure 2—figure supplement 2b*). Together, these results indicate that MORC2 forms biomolecular condensates under physiological conditions in vivo and in vitro.

## The MORC2 C-terminal domain (CTD) drives condensate formation

To identify the key determinants governing the assembly of MORC2 nuclear condensates, we partitioned the protein into two functional modules based on its domain architecture: the N-terminal domain (NTD, residues 1–472), comprising the ATPase and CC1 motifs, and the C-terminal domain (CTD, residues 473–1032), which encompasses the IDR and multiple coiled-coil regions. A series of EGFP-tagged truncation constructs were then expressed in HeLa cells to systematically map the regions essential for condensate formation (*Figure 3a*). Deletion of the CTD completely abolished puncta formation, resulting in a diffuse localization across the nucleus and cytoplasm (*Figure 3—figure supplement 1a*). By contrast, NTD deletion had negligible effects on nuclear localization or the capacity for condensation. To exclude the possibility that the NTD's diffuse pattern was merely due to its loss of nuclear entry, we appended an exogenous nuclear localization signal (NLS, PKKKRKV *Kalderon et al., 1984*) to the NTD; however, while this redirected the protein to the nucleus, it failed

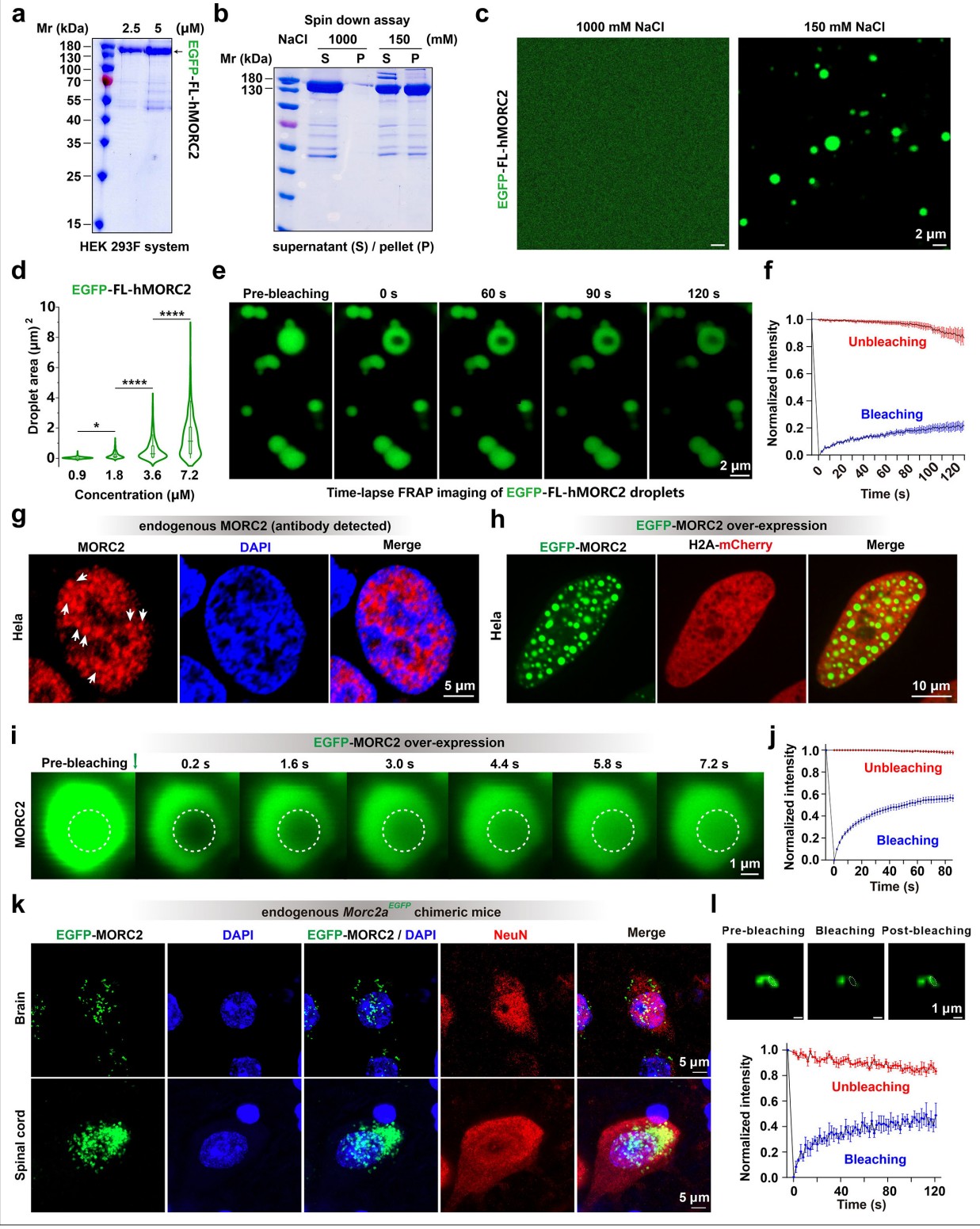

**Figure 2.** Endogenous MORC2 organizes into dynamic condensates within the nucleus. (**a**) SDS-PAGE analysis of EGFP-MORC2FL protein purified from HEK293F cells, showing protein concentration at 2.5 μM and 5 μM. (**b**) Sedimentation assay shows the distribution of EGFP-MORC2FL (7.2 μM) between supernatant (**S**) and pellet (**P**) fractions under 1000 mM (control) and 150 mM NaCl, indicating its propensity for phase separation. (**c**) EGFP-MORC2 (7.2 μM) undergoes phase separation into droplets in a buffer containing 150 mM NaCl, but not in 1000 mM NaCl buffer. Scaler bar: 2 μm. (**d**) The proteins were examined in a buffer containing 150 mM NaCl, and phase separation was assessed by fluorescence microscopy with 488 nm excitation for EGFP. Quantification of droplet areas from (*Figure 2—figure supplement 1a*), presented as violin plots with box plots indicating medians. Data

*Figure 2 continued on next page*

*Figure 2 continued*

are derived from ≥3 independent images and reflect the size distribution of phase-separated droplets. Data are presented as mean ± SEM; one-way ANOVA with Tukey's post hoc test. ****p<0.0001; *p<0.05. (**e**) Time-lapse FRAP imaging of EGFP-MORC2FL droplets formed in vitro, demonstrating fluorescence recovery over time. Scaler bar: 2 µm. (**f**) Quantitative analysis of fluorescence recovery from (**e**), showing the dynamic properties of in vitro MORC2 droplets. Data represent mean ± SEM, with n≥3 replicates. (**g**) Immunostaining of endogenous MORC2 in HeLa cells revealed punctate localized within the nucleus, likely representing condensates involved in transcriptional regulation (white arrows). Nuclei are counterstained with DAPI to visualize chromatin-rich regions. Scaler bar: 5 µm. (**h**) Live-cell imaging of transiently transfected EGFP-MORC2FL showed its assembly into dispersed, nearly spherical condensates in the nucleus, while stable expression of H2A-mCherry served to mark chromatin-enriched domains. Scaler bar: 10 µm. (**i**) Time-lapse FRAP analysis of EGFP-MORC2FL condensates in the nuclei of transiently transfected HeLa cells revealed rapid fluorescence recovery within seconds, indicative of their dynamic, nature. Scaler bar: 1 µm. (**j**) Quantitative analysis of fluorescence recovery from (**i**), showing the dynamic properties of in vitro MORC2 droplets. Data represent mean ± SEM, with n≥10 replicates. (**k**) Sections of brain and spinal cord from endogenous EGFP-MORC2 chimeric mice show EGFP-MORC2 condensation distribution in NeuN-positive neurons. Scaler bar: 10 µm. (**l**) Representative FRAP images from of FRAP experiments performed on fresh 250 µm brain slices of endogenous *Morc2a*$^{EGFP}$ chimeric mice demonstrate the dynamic nature of endogenous MORC2 condensates. Data are presented as mean ± SEM, with n=3 droplets analyzed. Scaler bar: 5 µm.

The online version of this article includes the following source data and figure supplement(s) for figure 2:

**Source data 1.** Original files for SDS-PAGE gel analysis displayed in *Figure 2a and b*.

**Source data 2.** PDF file containing original SDS-PAGE gel for *Figure 2a and b*, indicating the relevant bands and treatments.

**Figure supplement 1.** In vitro and in vivo evidence that MORC2 undergoes phase separation.

**Figure supplement 1—source data 1.** Original files for SDS-PAGE gel analysis displayed in *Figure 2—figure supplement 1b*.

**Figure supplement 1—source data 2.** PDF file containing original SDS-PAGE gel for *Figure 2—figure supplement 1b*, indicating the relevant bands and treatments.

**Figure supplement 2.** Genomic loci of *Morc2a* in chimeric mice and its subcellular localization during mitosis.

to restore puncta (*Figure 3a*), confirming that the CTD is the primary driver of MORC2 condensation. To assess whether the CTD is intrinsically prone to undergo condensation in vitro, we purified a CTDΔCW (residues 537–1032) variant from *Escherichia coli*, as the full CTD exhibited suboptimal biochemical stability (*Figure 3—figure supplement 2a, b*). At 150 mM NaCl, Cy3-labeled CTDΔCW readily formed discrete condensates in a concentration-dependent manner (*Figure 3b and c*), and displayed a sedimentation profile characteristic of phase-separated assemblies (*Figure 3d and e*). Fluorescence microscopy and SLS confirmed that Cy3 labeling did not perturb its condensation behavior, morphology, or molecular weight distribution (*Figure 3—figure supplement 2c–f*).

Further dissection of the CTD revealed that deletion of either the CC2 or TCD domains did not impair the formation of nuclear puncta (*Figure 3—figure supplement 1b*). In contrast, removal of the IDR-NLS region (residues 593–761) abolished both puncta formation and nuclear localization (*Figure 3—figure supplement 1b*). Notably, appending an exogenous NLS (PKKKRKV) to CTDΔIDR-NLS restored its nuclear entry but failed to rescue condensate assembly (*Figure 3—figure supplement 1c*), identifying the IDR as a prerequisite for condensation. Similarly, deletion of the CC3-IBD segment maintained nuclear localization but eliminated puncta formation, indicating that this region is also indispensable (*Figure 3—figure supplement 1b*). Intriguingly, neither the IDR-NLS fragment nor the CC3-IBD fragment alone (even when NLS-fused) was sufficient to trigger condensation (*Figure 3—figure supplement 1d*), implying a cooperative mechanism between these modules. Consistent with these cellular findings, in vitro assays showed that neither the MORC2ΔCC3-IBD (residues 1–900) nor the isolated CC3-IBD fragment could undergo condensation (*Figure 3—figure supplement 3a, b*). However, mixing the IDR with the IBD fragment reconstituted robust condensate formation at higher concentration (*Figure 3—figure supplement 3c*), confirming their synergistic role. Finally, the CW domain appears to modulate condensate morphology, as its deletion led to enlarged and frequently fused puncta. We conclude that the IDR, CC3, and IBD constitute the minimal structural core required for MORC2 condensation, while the CW domain serves to restrict condensate overgrowth and maintain proper morphology (*Figure 3a*, *Figure 3—figure supplement 1e*).

## Multivalent interactions between the IDR and IBD promote MORC2 condensation

Next, we elucidated the mechanistic basis for MORC2 assembly mediated by the IDR and CC3-IBD. We hypothesized that multivalent interactions between these modules drive the formation of

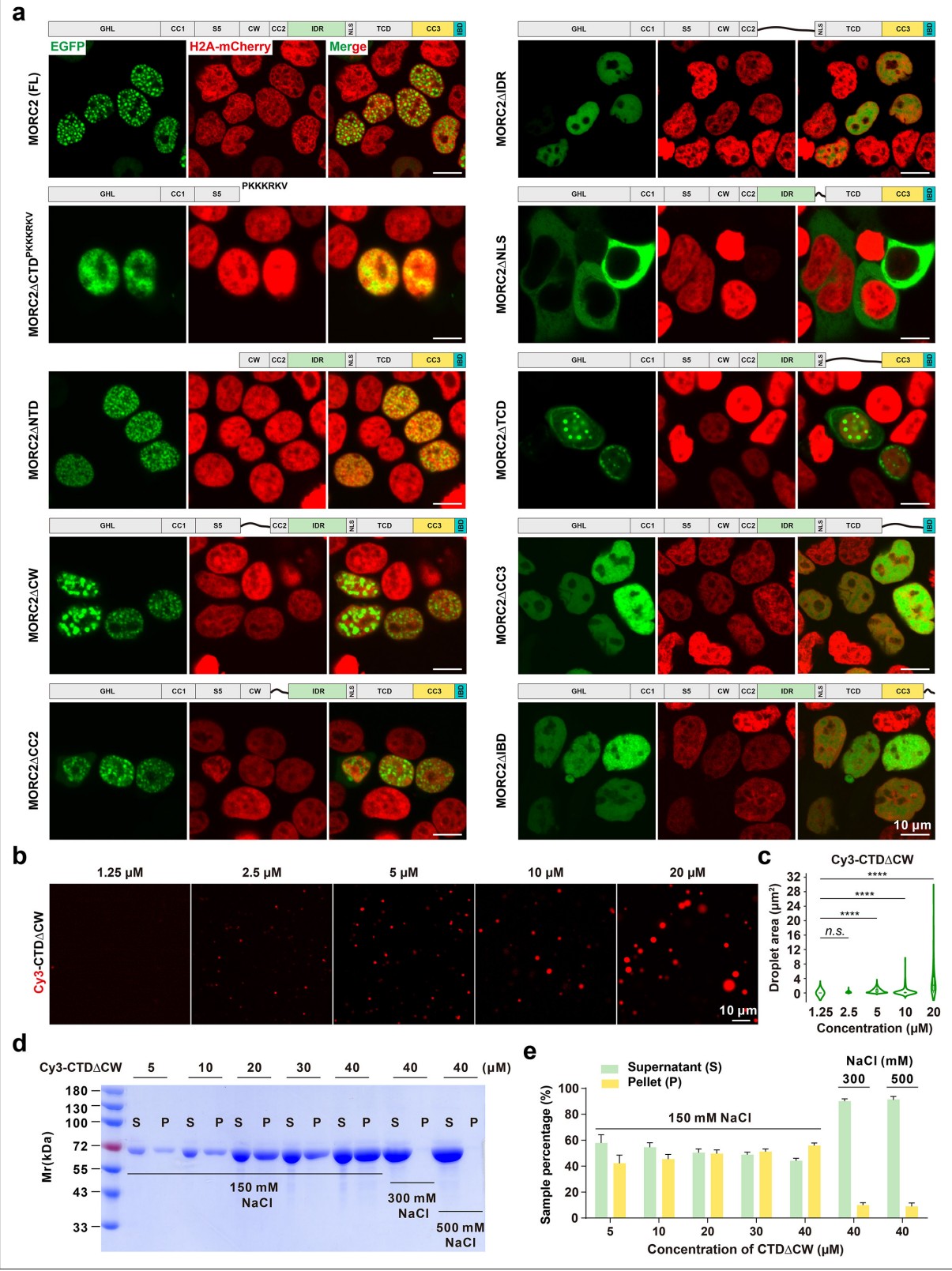

**Figure 3.** Multiple domains regulate phase separation of MORC2. (**a**) Sequential deletion analysis revealed that the NTD is dispensable for condensate formation, as EGFP-tagged NTD alone failed to form condensates and was excluded from the nucleus. Fusion with a canonical nuclear localization signal ('PKKKRKV') restored nuclear localization but did not rescue condensate formation. In contrast, the CTD alone was sufficient to form nuclear condensates. Deletion of the CW domain resulted in enlarged condensates in a subset of cells, while removal of CC2 or the TCD had no apparent

*Figure 3 continued on next page*

*Figure 3 continued*

effect on condensate formation. Strikingly, deletion of either the IDR or CC3 completely abolished nuclear condensate assembly. Deletion of the IBD markedly reduced the frequency of condensate formation. Loss of the intrinsic NLS led to cytoplasmic localization of MORC2; however, no condensates were detected in the cytoplasm under transient transfection conditions in HeLa cells. Together, these results define the IDR and CC3 as the minimal and essential elements for MORC2-mediated phase separation, and implicate the CW domain, IBD, and nuclear microenvironment in the fine-tuning of condensate assembly. Scaler bar: 10 µm. (**b**) In vitro phase separation of Cy3-labeled CTDΔCW at concentrations from 1.25 µM to 20 µM in 150 mM NaCl buffer, visualized as spherical condensates. Scaler bar: 10 µm. (**c**) Quantification of droplet areas from (**b**), displayed as violin plots with box plots indicating median values. Data are presented as mean ± SEM; one-way ANOVA with Tukey's post hoc test. ****$p < 0.0001$, *n.s.* not significant. (**d**) SDS-PAGE analysis of Cy3-CTDΔCW protein distribution between supernatant (**S**) and pellet (**P**) fractions after centrifugation at increasing concentrations (5 µM to 40 µM) in 150 mM NaCl buffer. At 40 µM, increasing the NaCl concentration to 300 mM or 500 mM reduced droplet formation, indicating salt sensitivity. (**e**) Quantification of Cy3-CTDΔCW showing the percentage of soluble protein from densitometric analysis of S/P fractions in panel (**d**), based on three independent biological replicates.

The online version of this article includes the following source data and figure supplement(s) for figure 3:

**Source data 1.** Original files for SDS-PAGE gel analysis displayed in *Figure 3d* and corresponding to *Figure 3e*.

**Source data 2.** PDF file containing original SDS-PAGE gel for *Figure 3d*, indicating the relevant bands and treatments.

**Figure supplement 1.** The C-terminal domain (CTD) of MORC2 mediates nuclear condensate formation.

**Figure supplement 2.** The biophysical behavior of CTD and CTDΔCW in vitro.

**Figure supplement 3.** In vitro assays to investigate isolated fragment fails to induce phase separation.

**Figure supplement 3—source data 1.** Original files for SDS-PAGE gel analysis displayed in *Figure 3—figure supplement 3a*.

**Figure supplement 3—source data 2.** PDF file containing original SDS-PAGE gel for *Figure 3—figure supplement 3a*, indicating the relevant bands and treatments.

a condensed phase that coexists with a dilute aqueous phase. Using nuclear magnetic resonance (NMR) spectroscopy, we characterized the interaction between the IDR and the C-terminal segment (residues 901–1032). $^{15}$N-HSQC titration assays using the $^{15}$N-labeled CC3 (residues 901–1003) or the 1004–1032 fragment revealed that residues 1004–1032 constitute the primary binding site for the IDR, which we termed the IDR-Binding Domain (IBD), whereas the CC3 region exhibited negligible interaction (*Figure 4a–c*).

Further subdivision of the IDR into three segments, IDRa (residues 593–643), IDRb (residues 644–694), and IDRc (residues 695–735), pinpointed IDRa as the key region responsible for interaction with IBD (*Figure 4a and d–f*). As is typical for IDR-mediated interactions involved in condensation, the observed chemical shift perturbations (CSPs) were relatively small but highly region-specific. Notably, the CSP profile of IDRa closely mirrors that of the full-length IDR, whereas IDRb and IDRc showed minimal perturbations (*Figure 4—figure supplement 1*). Detailed sequence analysis revealed that the CSPs are clustered around charged residues (D, E, K, R) and their neighboring residues (Y, T, N, I, L, P). This suggests that the association is driven by a concerted action of multiple non-covalent forces: beyond the foundational electrostatic attraction, the involvement of aromatic (Y) and aliphatic (I, L, P) residues points to the contribution of π-π stacking, cation-π interactions, and hydrophobic effects, while polar residues (T, N) likely facilitate intermolecular hydrogen bonding. This hierarchical assembly, characterized by both charge complementarity and hydrophobic-like contributions, aligns with the 'sticker-and-spacer' model. In this framework, these multivalent residues act as 'stickers' that provide the requisite valency to scaffold a robust yet dynamic network, while the intervening sequences serve as 'spacers' to maintain the fluidity necessary for the condensed state (*Figure 4g*). Collectively, these quantitative CSP data support a model of weak, dynamic, and region-specific interactions, a hallmark of IDR-driven condensation mechanisms (*Alderson and Kay, 2021*; *Kim et al., 2019*; *Murthy et al., 2019*).

To assess the role of IDRa-IBD multivalent interactions in cells, we transfected HeLa and HEK293T cells with MORC2FL variants harboring individual deletions of the IBD, IDRa, IDRb, or IDRc regions. Deletion of the IBD abolished condensate formation in both HeLa cells (*Figure 3a*, *Figure 3—figure supplement 1e*) and HEK293T cells (*Figure 4—figure supplement 2*). Similarly, deletion of IDRa significantly impaired condensation, resulting in a marked decrease in both the number and area of nuclear puncta, whereas IDRb or IDRc deletions had minimal impact (*Figure 4h and i*). Based on these findings, we propose a model in which multivalent 'sticker' interactions between IDRa and the IBD

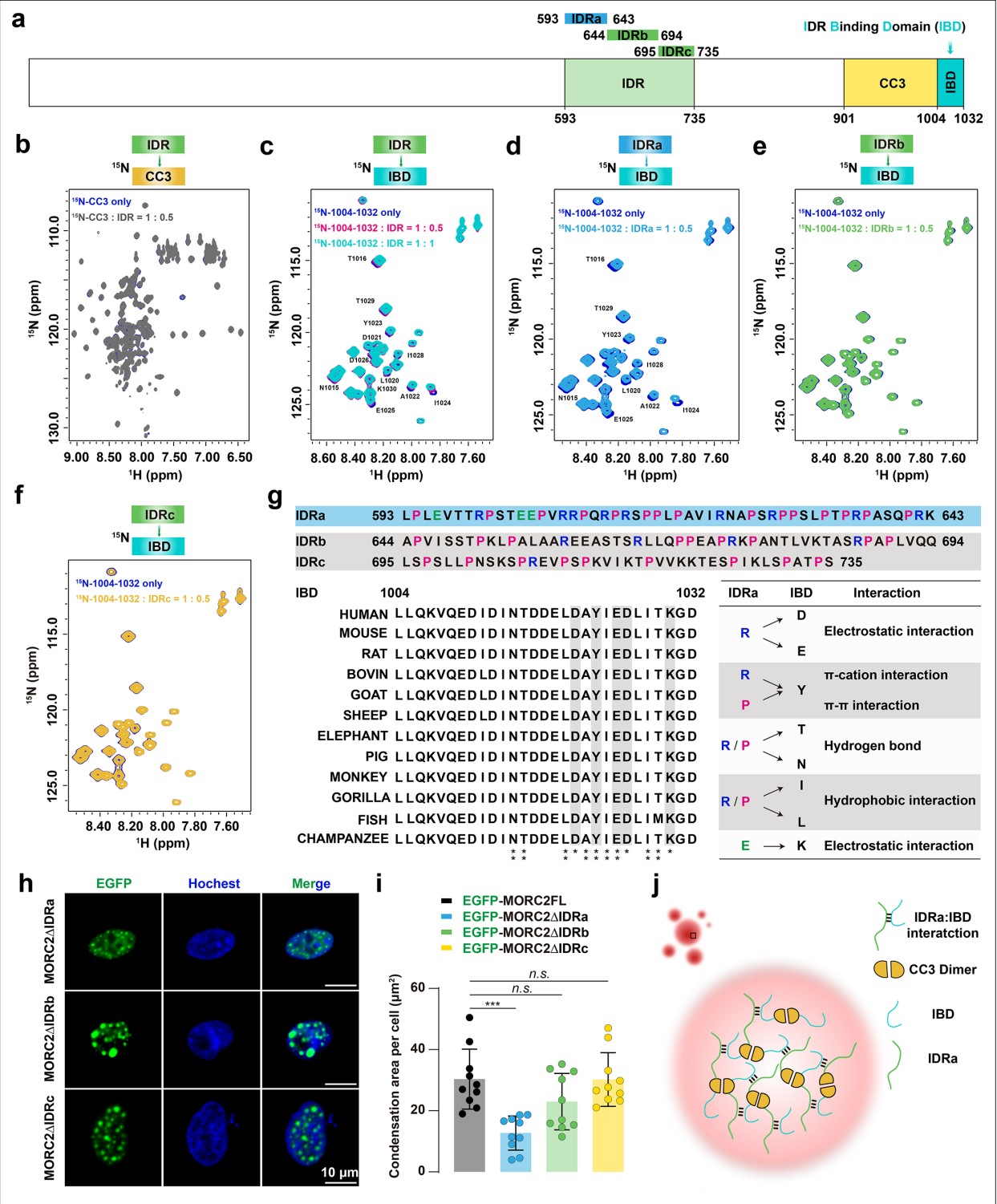

**Figure 4.** Multivalent interactions between IDR and IBD drive MORC2 condensation. (**a**) Domain organization of MORC2 C-terminal regions analyzed by NMR titration, including the IDR, CC3, and IBD. (**b**) $^{15}$N-labeled CC3 (residues 901–1003) shows no chemical shift upon titration with IDR (residues 593–735), indicating no detectable interaction. (**c**) $^{15}$N-labeled IBD (residues 1004–1032) exhibits clear chemical shift perturbations (CSPs) upon IDR titration, confirming a direct, specific interaction. (**d-f**) NMR titrations of $^{15}$N-labeled IBD with IDR subregions: IDRa (593-643), IDRb (644-694), and IDRc (695-735). IDRa induces clear CSPs (**d**), while IDRb (**e**) and IDRc (**f**) show minimal or no shifts, identifying IDRa as the primary IBD-binding segment. (**g**) Sequence analysis of IDR subregions reveals that IDRa is enriched in proline and arginine residues, enabling electrostatic interactions. Sequence alignment of IBD across species highlights conserved residues involved in IDRa binding. A summary table lists representative residues from IDRa and IBD exhibiting CSPs

*Figure 4 continued on next page*

*Figure 4 continued*

in titration assays (marked with asterisks). (**h**) Representative confocal images of HeLa cells expressing EGFP-tagged MORC2ΔIDRa, ΔIDRb, or ΔIDRc constructs. Scale bar: 10 μm. (**i**) Quantification of droplet area per cell in (**h**). Deletion of IDRa significantly impairs condensate formation. n=10 cells per condition. Data are presented as mean ± SEM; statistical analysis: one-way ANOVA with Tukey's post hoc test. \*\*\*p<0.001; *n.s.* not significant. (**j**) Working model of MORC2 condensates. CC3 dimerization serves as a structural scaffold, while weak, transient, but specific multivalent interactions between IDRa and IBD cooperatively promote condensate formation.

The online version of this article includes the following figure supplement(s) for figure 4:

**Figure supplement 1.** NMR titration about IDR-IBD.

**Figure supplement 2.** IBD-deficient mutant (ΔIBD) in HEK293T cells.

drive MORC2 intermolecular assembly, a process further potentiated by CC3-mediated dimerization, which collectively provides the valency threshold required for condensation (*Figure 4j*).

## DNA binding promotes MORC2 condensation and stimulates ATPase activity

Notably, MORC2 condensates form exclusively within the nuclei of transfected HeLa cells, implying that chromatin-associated factors, most likely DNA, serve as essential cues for their nucleation or stabilization. While the N-terminal CC1 domain was previously identified as a DNA-binding module (*Douse et al., 2018*), we hypothesized that the full-length MORC2 architecture utilizes additional interfaces to facilitate complex chromatin interactions.

To test this, we performed a comprehensive domain screen using electrophoretic mobility shift assays (EMSA). Beyond confirming the robust DNA-binding activity of the CC1 domain (*Figure 5a and b*), our screen revealed that the CC2, IDR, and TCD (Tudor Chromo-like domain) also possess significant DNA-binding capabilities (*Figure 5c–e*). In contrast, the CC3–IBD fragment showed no affinity for DNA (*Figure 5f*). These results indicate that MORC2 employs a multivalent DNA-binding architecture distributed across both its N-terminal and C-terminal regions.

Building on this, we investigated whether these multivalent DNA-protein interactions could scaffold MORC2 condensation in vitro. At 0.9 μM, MORC2FL alone failed to form discrete assemblies; however, the addition of 25 nM 601 DNA induced robust condensate formation, demonstrating that DNA acts as a molecular seed to promote MORC2 phase separation (*Figure 5g and h*). Interestingly, while the NTD (residues 1–536) binds DNA, it remains diffuse (*Figure 3—figure supplement 3d, e*), whereas the CTDΔCW (residues 537–1032) undergoes DNA-driven condensation (*Figure 5i*). Systematic evaluation across a matrix of protein and DNA concentrations yielded a phase diagram mapping the transitions between clear solutions, liquid-like condensates, and amorphous precipitates, consistently showing that DNA significantly lowers the concentration threshold for MORC2 condensation (*Figure 5—figure supplement 1*). This functional divergence suggests a division of labor within MORC2: while multiple domains provide DNA affinity, the CTD provides the essential protein-protein interaction network required for higher-order assembly.

Finally, we explored how this assembly impacts the MORC2 enzymatic core. The addition of 601 DNA markedly stimulated the ATPase activity of wild-type MORC2, but not the ATP-binding-deficient N39A mutant (*Figure 5j*). This reveals a synergistic interplay: DNA binding does not merely recruit MORC2 to chromatin; it acts as a molecular scaffold that facilitates condensation, which in turn may optimize the productive conformation or local density required for allosteric activation of the ATPase domain. In conclusion, these findings highlight a coordinated mechanism that integrates DNA binding, condensation, and enzymatic activity, providing a biophysical basis for the multivalent assembly and allosteric regulation of MORC2 on chromatin.

## Condensate formation and dynamics are required for MORC2-mediated transcriptional regulation

Establishing a causal link between biomolecular condensation and biological function, particularly in transcriptional regulation, remains a central challenge. A key question is whether MORC2 condensates represent discrete functional entities or merely inactive byproducts. To address this, we sought to decouple condensate formation from internal dynamics, allowing for a rigorous interrogation of their functional relevance. While deletion of the CC3 domain abolishes MORC2 condensation, CC3

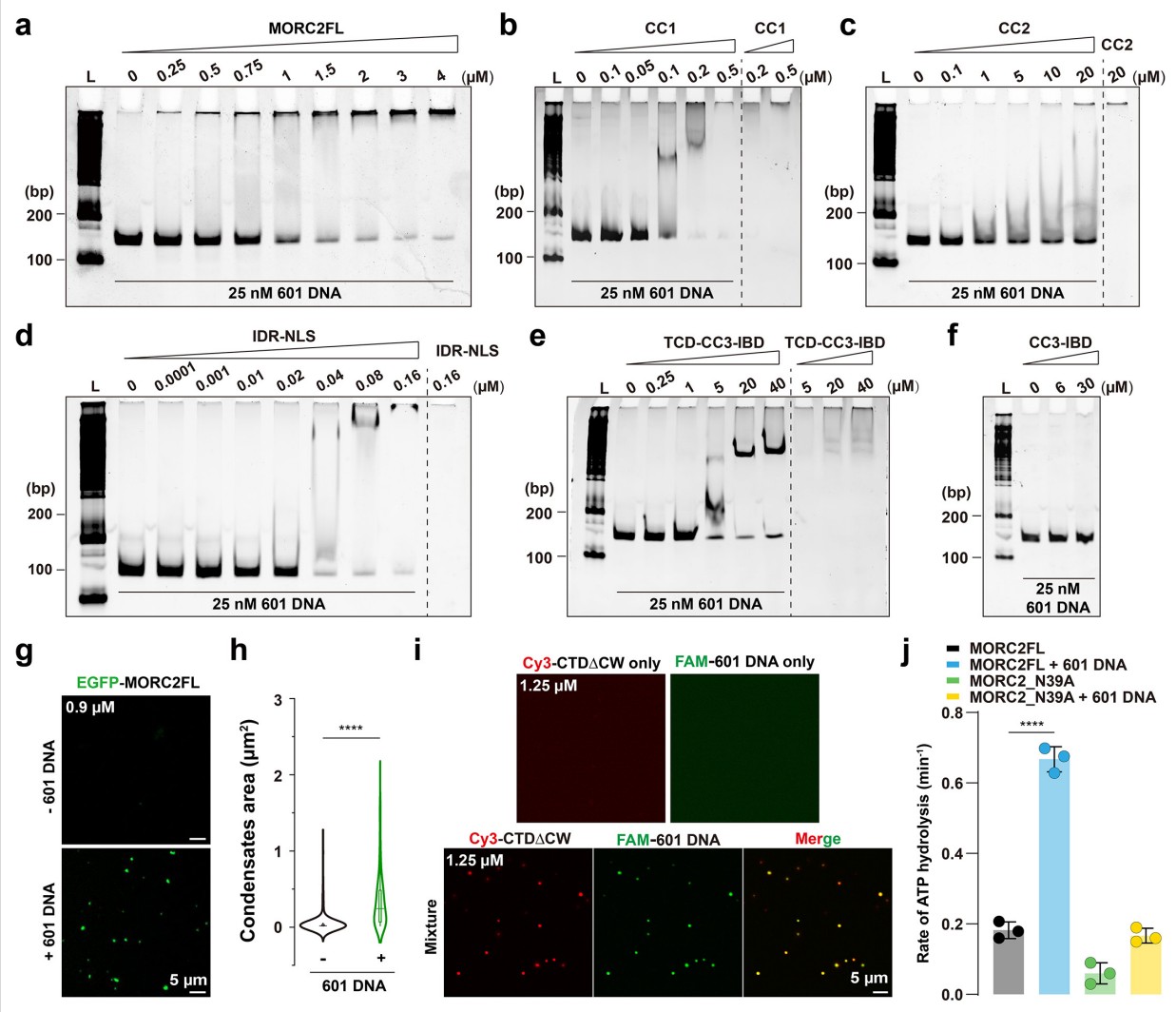

**Figure 5.** DNA binding enhances MORC2 condensation and modulates its ATPase activity. (**a**) Electrophoretic mobility shift assay (EMSA) showing that MORC2FL binds 25 nM 601 DNA, as evidenced by shifts in DNA mobility corresponding to protein-DNA complexes. (**b**) EMSA highlighting the strong DNA-binding affinity of the CC1 domain, corroborating previous reports that identify this region as a key mediator of DNA interaction. (**c–f**) EMSA results demonstrating the binding capacities of individual MORC2 domains to 25 nM 601 DNA: CC2 (**c**), IDR (**d**), TCD-CC3-IBD (**e**), and CC3-IBD (**f**). The IDR exhibits robust DNA-binding activity, CC2 and TCD-CC3-IBD show weaker interactions, and CC3-IBD alone fails to bind DNA. These findings identify CC1, CC2, IDR, and the TCD as DNA-binding domains of MORC2. (**g**) Confocal imaging showing that 601 DNA promotes MORC2 phase separation. At 0.9 μM, EGFP-MORC2FL alone does not form droplets, but the addition of 25 nM 601 DNA induces distinct condensate formation. Scaler bar: 5 μm. (**h**) Quantification of MORC2 condensate sizes with or without 601 DNA. Violin plots illustrate the distribution of droplet sizes, with box plots indicating median values. Data are presented as mean ± SEM; unpaired student $t$-test. ****$p < 0.0001$. (**i**) Fluorescence microscopy showing Cy3-labeled CTDΔCW (1.25 μM) coalescing with 25 nM FAM-labeled 601 DNA. DNA is recruited into MORC2 droplets, forming distinct condensates. Scaler bar: 5 μm. (**j**) In the presence of 10 nM 601 DNA, the ATP hydrolysis activity of MORC2FL is significantly enhanced. The N39A mutant, which is deficient in ATP binding, serves as a negative control; nevertheless, a slight but detectable increase in ATPase activity is observed upon DNA binding. ATP hydrolysis activity of MORC2FL is significantly increased in the presence of 10 nM 601 DNA. The N39A mutant, which is deficient in ATP binding, serves as a negative control. Data are presented as mean ± SEM; statistical analysis: one-way ANOVA with Tukey's post hoc test. ****$p < 0.0001$.

The online version of this article includes the following source data and figure supplement(s) for figure 5:

**Source data 1.** Original files for EMSA analysis displayed in *Figure 5a-f*.

**Source data 2.** PDF file containing original EMSA images for *Figure 5a-f*, indicating the relevant bands and conditions.

**Figure supplement 1.** Phase diagram of MORC2-DNA.

is also indispensable for dimerization, confounding functional interpretations. Furthermore, although deleting IDRa markedly weakens MORC2 condensation, a tool to specifically manipulate the solubility and internal mobility of these assemblies was previously lacking.

To overcome these limitations, we employed a micropeptide-based 'killswitch (KS) strategy fused to the C-terminus of MORC2 (*Figure 6a*). This approach selectively reduces condensate dynamics without disrupting protein expression, folding, or domain architecture (*Zhang et al., 2025*). Unlike the ΔCC3 or ΔIDRa mutants, the MORC2 +KS variant robustly form nuclear condensates but exhibits markedly reduced internal dynamics, as demonstrated by FRAP analyses showing minimal fluorescence recovery (*Figure 6b and c*). This strategy allowed us to perturb the material properties of the condensates independently of structural integrity.

To evaluate how the modulation of MORC2 condensates influences its gene regulatory capacity, we compared the transcriptional consequences of rescuing *MORC2*-knockout HeLa cells with MORC2FL, condensation-deficient mutants (ΔCC3 and ΔIDRa), and the dynamics-defective MORC2 +KS (*Figure 6d*). Bulk RNA-seq was performed 48 hr post-transfection. We hypothesized that impairing either the assembly or the fluidity of the condensates would compromise MORC2-dependent gene regulation. Notably, despite being expressed at substantially higher levels than MORC2FL (*Figure 6e*), all three mutants showed a striking and consistent failure to restore the MORC2-dependent transcriptional profile (*Figure 6f–h*). This defect was particularly pronounced for transcriptionally repressed genes, including two sets of high-confidence MORC2 targets identified in prior studies (*Tchasovnikarova et al., 2017*; *Fendler et al., 2025*; *Figure 6i*, and *Figure 6—figure supplement 1*). Gene Ontology analysis revealed that the MORC2-regulated genes are involved in DNA damage responses, viral infection, and cell cycle regulation (*Figure 6g*). These findings demonstrate that neither increased protein abundance nor the mere presence of static, condensate-like structures is sufficient to restore MORC2 function.

Collectively, these data support a model in which the dynamic nature of MORC2 condensates is essential for full transcriptional activity. While soluble MORC2 complexes are likely involved in initial target recognition, our results indicate that proper condensate formation, and critically, internal fluidity, are required for effective transcriptional control. The inability of the MORC2 +KS mutant to rescue transcriptional defects, despite forming visible nuclear assemblies, strongly argues against a model where MORC2 condensates are merely inert byproducts of its activity.

## Pathogenic MORC2 variants differentially modulate condensation, DNA binding, and ATPase activity

MORC2 is essential for gene silencing at H3K9me3-marked loci and retrotransposon repression (*Müller and Helin, 2024*; *Pandiloski et al., 2024*; *Fukuda et al., 2018*; *Liu et al., 2018*). Pathogenic mutations in its GHL-type ATPase domain have been linked to severe neuropathies and cancer (*Jacquier et al., 2022*; *Stafki et al., 2023*; *Chung et al., 2024*; *Ding et al., 2018*; *Zhang et al., 2024*). To explore whether these mutations influence the condensation behavior and biochemical activities of the full-length protein, we examined a panel of disease-associated variants, including those linked to Charcot–Marie–Tooth disease type 2Z (CMT2Z: E236G, R252W, Q400R, D466N) and spinal muscular atrophy (SMA: S87L, S218L, F256L, R266A, T424R; *Figure 7a*).

In HeLa cells, all variants retained the ability to form nuclear puncta. However, the CMT2Z-linked E236G and, notably, the SMA-linked T424R mutants significantly increased the proportion of cells exhibiting nuclear condensates (*Figure 7b*). These observations suggest that the N-terminal region plays a nuanced role in modulating the phase boundary of MORC2. Specifically, previous structural studies indicated that the T424R mutation has minimal effects on the overall N-terminal architecture but primarily alters the assembly and dissociation kinetics of the N-terminal dimer (*Douse et al., 2018*). This suggests that the conformational cycling or dimerization state of the N-terminus directly contributes to the macroscopic regulation of MORC2 condensates. Interestingly, while deletion of the N-terminal region did not abolish puncta formation, FRAP analyses revealed a marked loss of condensate fluidity compared with the full-length protein (*Figure 7—figure supplement 1*). This highlights the N-terminus as a critical regulator of the material properties of MORC2 assemblies. Consequently, we evaluated the dynamics of E236G and T424R condensates; FRAP measurements showed substantially reduced recovery for E236G, indicating decreased internal fluidity, whereas T424R exhibited minimal effects (*Figure 7e and f*).

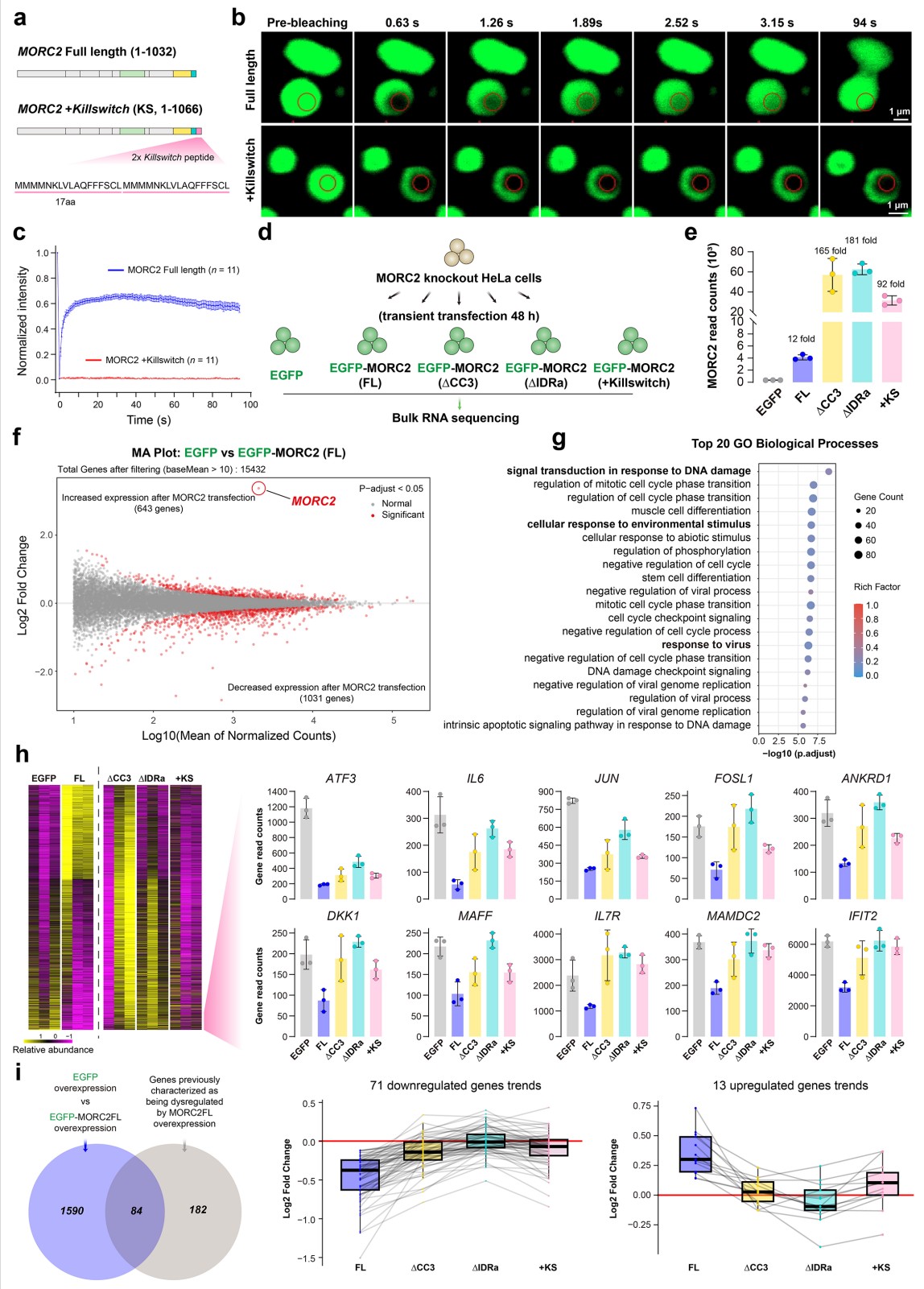

**Figure 6.** Dynamic MORC2 condensation is required for its transcriptional regulatory activity. (**a**) Schematic diagram showing the fusion of two 17–amino acid killswitch sequences to the C terminus of MORC2. (**b**) Live-cell FRAP images of HeLa cells overexpressing EGFP-MORC2 full length (FL) and EGFP–MORC-Killswitch (+KS). Scaler bar: 1 μm. (**c**) FRAP analysis of EGFP-MORC2 condensates in the nucleus. Data represent mean ± SEM, with n=11 replicates. (**d**) Schematic of the rescue experiment strategy. HeLa cells with CRISPR-Cas9-mediated *MORC2*-knockout were reconstituted with EGFP,

*Figure 6 continued on next page*

*Figure 6 continued*

EGFP-MORC2FL, or phase separation-deficient mutants EGFP-MORC2ΔCC3 and EGFP-MORC2ΔIDR, and dynamics-defective+KS mutant. (**e**) RNA-seq-derived read counts for the MORC2 gene in EGFP, EGFP-MORC2FL, EGFP-MORC2ΔCC3, EGFP-MORC2ΔIDR, and EGFP-MORC2FL +KS expressing cells. (**f**) The effect of MORC2 rescue on the transcriptome. MA plot showing all significantly differentially expressed genes between EGFP-MORC2FL and EGFP transfection in *MORC2*-knockout HeLa cells (n=1,674; DESeq2 adjusted p<0.05), highlighted in red. (**g**) GO-based biological process terms enriched among all differentially expressed genes. Data represent mean ± SD, with n=3 replicates. (**h**) Abundance trajectories of all regulated genes, normalized to their respective mean values. Shown is a selected panel of ten downregulated genes that are influenced by MORC2 rescue but show reduced responsiveness to all three mutants. (**i**) Venn diagram illustrating the overlap between differentially expressed genes identified following EGFP-MORC2FL overexpression in this study and previously characterized MORC2-regulated targets. Fold-change (log₂ fold change) analysis of the 84 overlapping genes showing significant expression changes upon MORC2FL overexpression, compared with their corresponding fold changes following overexpression of the three MORC2 mutants. Dashed lines connect the same gene across the four transfection conditions.

The online version of this article includes the following figure supplement(s) for figure 6:

**Figure supplement 1.** Condensation and dynamics-defective MORC2 mutants impair transcriptional regulation.

In vitro ATPase assays revealed that T424R significantly increases enzymatic activity (*Figure 7c*), consistent with its reported role in promoting ATPase dimer cycling (*Douse et al., 2018*). Additionally, S218L and F256L variants enhanced ATP hydrolysis, whereas others, such as Q400R and D466N, showed activity comparable to WT (*Figure 7c*). Due to severe solubility issues, E236G could not be biochemically characterized in vitro (*Figure 7—figure supplement 2e*).

We next assessed DNA binding affinity of MORC2 variants using fluorescence polarization (FP) assays. Despite subtle differences, none of the variants showed statistically significant changes in DNA-binding capacity (*Figure 7d*), indicating that these mutations primarily perturb enzymatic turnover and condensation dynamics rather than DNA recruitment.

In summary, our results support a model in which MORC2 mediates transcriptional regulation through its intrinsic ability to undergo biomolecular condensation. The N-terminal ATPase domain undergoes conformational cycling between open and closed states, while the C-terminal CC3-mediated dimer acts as a stable scaffold for condensate formation (*Figure 7g*). MORC2 condensation is further fine-tuned by multivalent interactions between the IBD and IDRa regions, as well as by DNA binding. Together, these features enable MORC2 to bridge distal chromatin regions and establish repressive condensates, underscoring its central role in chromatin organization and genome stability.

## Discussion

### Biophysical principles of MORC2 organization

While the MORC family of ATPases is recognized as a central player in epigenetic regulation (*Li et al., 2013*), genome stability (*Koch et al., 2017*; *Dong et al., 2018*), and meiotic progression (*Inoue et al., 1999*; *Wang et al., 2010*), the biophysical principles governing their spatiotemporal organization have remained elusive. In this study, we establish a causal link between MORC2 condensation and its transcriptional regulatory function. By integrating high-resolution structural insights with functional assays, we reveal that MORC2 does not act merely as a solitary chromatin remodeler (*Downs and Gasser, 2024*; *Li et al., 2012*; *Tan et al., 2025*); instead, it functions as phase separation-driven DNA compaction machinery. This discovery demonstrates how mesoscale material properties, specifically the fluidity and density of protein-DNA assemblies, are directly translated into precise gene regulation.

### A modular engine for condensate assembly

The architectural basis of MORC2 assembly relies on a finely tuned multi-domain interplay. The C-terminal region, comprising the structured CC3-IBD and the intrinsically disordered IDR, provides the mechanistic engine for this process. CC3 serves as a critical structural scaffold (*Figures 1e and 3a*), mediating homodimerization that effectively doubles the molecular valency, thereby facilitating multivalent 'sticker' interactions between the IDR and the defined IBD (*Figure 4*). We found that mutations selectively abrogating condensate formation, or those that arrest condensate dynamics without blocking assembly, impair MORC2's ability to repress core target genes (*Figure 6*). This underscores that high internal fluidity is functionally paramount, ensuring the rapid exchange kinetics required for continuous ATP hydrolysis and dynamic chromatin engagement.

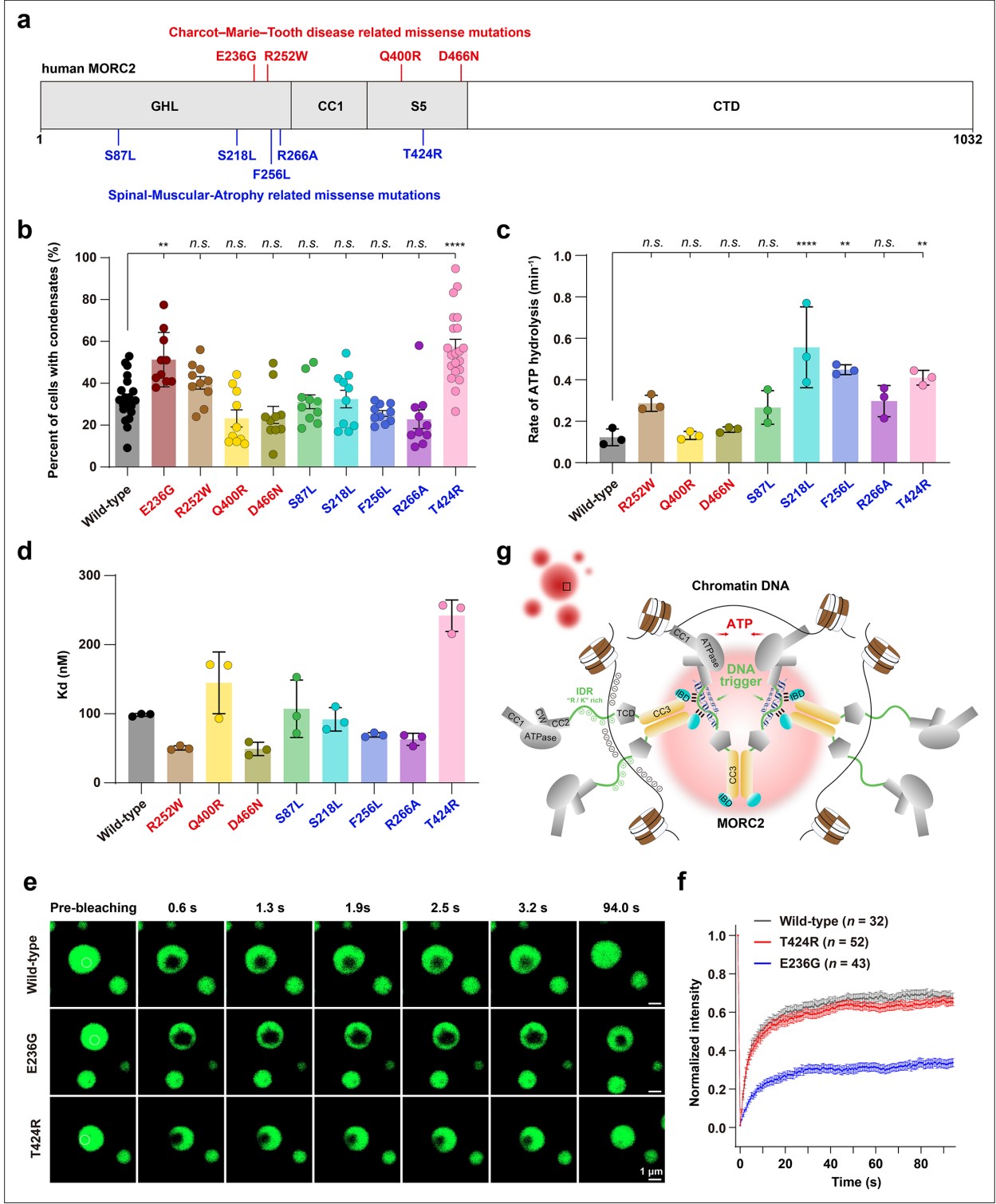

**Figure 7.** Pathogenic variants of MORC2 alter condensates behavior, DNA binding, and ATPase activity. (**a**) Schematic illustration of pathogenic MORC2 mutations associated with Charcot-Marie-Tooth disease type 2Z (CMT2Z, red) and spinal muscular atrophy (SMA, blue), mapped onto the MORC2 domain structure. (**b**) Quantification of HeLa cells transfected with WT MORC2 or nine pathogenic mutations exhibiting nuclear condensates was performed for each field of view. E236G and T424R show the most pronounced enhancement of phase separation. n≥10 fields. Data are presented as mean ± SEM; one-way ANOVA with Tukey's post hoc test. ****p < 0.0001, **p < 0.01, *n.s.* not significant. (**c**) ATPase activity of WT MORC2 and eight pathogenic variants, measured under revised conditions. S218L, F256L, and T424R exhibit significantly elevated activity. We were unable to purify MORC2 constructs bearing the E236G mutation from either HEK293F cells, suggesting that it may cause misfolding of the ATPase module. Data are

*Figure 7 continued on next page*

*Figure 7 continued*

presented as mean ± SEM; one-way ANOVA with Tukey's post hoc test. ****$p < 0.0001$, **$p < 0.01$, *n.s.* not significant. (**d**) Fluorescence polarization (FP) analysis of DNA binding affinities to the 601 DNA sequence. All protein variants were evaluated under uniform assay conditions, including those pertinent to phase separation. Dissociation constants ($K_d$) were determined by fitting the FP data to a log(agonist) vs. response model and are reported as mean ± SEM: WT (98±1 nM), R252W (51±2 nM), Q400R (145±26 nM), D466N (49±6 nM), S87L (107±24 nM), S218L (92±10 nM), F256L (69±2 nM), R266A (63±5 nM), and T424R (242±13 nM). Several variants, including R252W, D466N, F256L, and R266A, demonstrated increased binding affinity relative to WT, while others, such as T424R and Q400R, exhibited markedly reduced binding. (**e**). Live-cell FRAP images of HeLa cells overexpressing EGFP-MORC2 full length (FL), E236G, and T424R mutants. Scaler bar: 1 μm. (**f**). FRAP analysis of EGFP-MORC2 condensates in the nucleus. Data represent mean ± SEM, with n≥32 replicates. (**g**) A mechanistic model of MORC2-mediated chromatin condensation. MORC2 functions as a multivalent DNA-binding machine. The C-terminal CC3-IBD and IDR serve as the structural engine for phase separation, while DNA acts as a scaffold that lowers the phase boundary. The N-terminal ATPase domain undergoes conformational cycling (Open/Closed), providing the dynamic fluidity necessary for the assembly to remain within a functional 'viscoelastic window'. In this window, MORC2 achieves the local density required for DNA compaction and gene silencing.

The online version of this article includes the following figure supplement(s) for figure 7:

**Figure supplement 1.** Compared to the wild-type protein, N-terminally truncated MORC2-CTD forms nuclear puncta in HeLa cells with significantly reduced internal dynamics, as revealed by FRAP assays.

**Figure supplement 2.** Representative SDS-PAGE gels for all purified protein constructs.

## The 'viscoelastic window' and molecular stoichiometry

A defining insight from our study is that the mere presence of nuclear puncta is insufficient for MORC2's repressor activity. Instead, its function is governed by the precise material state of the condensates. We demonstrate that DNA acts as an essential functional scaffold (*Figure 5*), driving condensation potentially mediated by nonspecific electrostatic interactions with the arginine/lysine-rich IDR. Crucially, these assemblies are exquisitely sensitive to molecular stoichiometry. This establishes a narrow, functionally permissive 'viscoelastic window': Too diffuse: MORC2 fails to achieve the local density necessary for mechanical DNA compaction. Too rigid: If the system crosses the threshold into a solid-like, arrested state (as seen in the MORC2 +KS mutant or E236G variant), it loses the fluidity required for enzymatic turnover and co-repressor engagement. Within the nucleus, the CW domain likely serves as a regulatory node, orchestrating the compositional regulation of these condensates by reading specific epigenetic marks to fine-tune local valency and morphology (*Figure 3*, *Figure 3—figure supplement 1*). This concept resonates with findings in bacterial condensate biology: in *Caulobacter crescentus*, the PopZ condensate similarly requires an optimal liquidity for proper cell fate determination, with both hyper-fluid and excessively rigid PopZ assemblies being deleterious to bacterial fitness (*Lasker et al., 2022*). Our findings extend this principle to a chromatin-regulatory context in metazoans, suggesting that a permissive 'viscoelastic window' may be a conserved design principle for functional condensates across kingdoms of life.

## Mechanistic insights into pathogenic variants

Our analysis of disease-associated variants, particularly E236G (CMT2Z) and T424R (SMA), highlights how mutations in the ATPase domain can pleiotropically affect both enzymatic turnover and phase behavior (*Figure 7*). While T424R primarily accelerates the ATPase dimer cycle (*Douse et al., 2018*), E236G triggers a transition toward a less dynamic, 'hardened' condensate state. These results emphasize that disease mutations do not simply 'turn off' the protein; they perturb the biophysical context of the assembly. The observation that some variants exhibit elevated ATPase activity in vitro suggests that in the native cellular environment, condensation-mediated local enrichment may be required to properly couple ATP hydrolysis to productive chromatin remodeling.

## Interpretive considerations: condensate functionality versus sequestration

While our killswitch (KS) experiments provide compelling evidence that the dynamic material properties of MORC2 condensates are required for transcriptional repression, an important alternative interpretation warrants explicit acknowledgement. The KS peptide enhances homotypic protein-protein interactions within the condensate, which in principle could increase the partitioning coefficient of MORC2 +KS into the condensate relative to wild-type MORC2. If so, the transcriptional defect observed upon MORC2 +KS expression might not reflect a loss of function within the condensates,

but rather a depletion of the functionally active, dilute-phase MORC2 pool through sequestration, a mechanism conceptually parallel to that proposed for certain transcription factors (*Chong et al., 2022*). We note that the confocal images in *Figures 2h and 3a* reveal a visible diffuse nuclear signal in addition to puncta for wild-type MORC2, and a careful quantitative comparison of this extracondensate fraction between MORC2FL and MORC2 +KS would be critical to distinguish these two models. If the proportion of diffuse MORC2 +KS outside condensates is similar to that of wild-type MORC2, sequestration would be an unlikely explanation for the observed functional deficit. Although generating such definitive data remains technically challenging in a transient transfection system, we acknowledge this as a limitation of the current study and emphasize that our data are most consistent with, rather than formally proving, a model in which function resides inside dynamic MORC2 condensates.

A related caveat concerns the inter-replicate variability observed in the RNA-seq data, particularly for the MORC2ΔCC3 mutant (*Figure 6h*). We attribute this variability primarily to the stochastic nature of transient transfection, which inherently produces heterogeneous expression levels across cells and replicates. While the transcriptional phenotypes of specific high-confidence MORC2 target genes (*Figure 6i*) are consistent across replicates and corroborate our main conclusions, the global transcriptomic landscape, as depicted in the heatmap of *Figure 6h*, exhibits notable inter-replicate scatter, especially for condensate-deficient mutants. We thus acknowledge that the RNA-seq data, as currently presented, provides strong but not irrefutable support for our central claim. Importantly, despite their divergent structural consequences, all three mechanistically distinct perturbations, ΔCC3, ΔIDRa, and +KS, converge on a strikingly consistent failure to restore transcriptional regulation of previously validated MORC2 target genes. This phenotypic convergence across orthogonal modes of condensate disruption provides strong internal coherence to our interpretation, even in the context of transcriptome-wide variability at the global level.

Taken together, these considerations reinforce our working model while highlighting the need for future studies, ideally using stable knock-in systems and single-cell transcriptomics, to resolve whether the dilute-phase or condensed-phase MORC2 is the primary functional species.

Our findings clarify previous discrepancies regarding DNA-binding properties of MORC2. By utilizing the 601 nucleosome-positioning sequence, we demonstrate a much more robust interaction than previously captured with short, random dsDNA. Furthermore, the use of full-length MORC2 reveals that C-terminal regulatory domains are essential for maintaining enzymatic homeostasis, a feature masked in truncated constructs. Whether *MORC2*-mediated silencing is achieved through direct modulation of ATPase cycles or an independent mechanical compaction mechanism remains an open question.

While our study establishes the essential role of MORC2 condensation in transcriptional control, this exploration represents an initial step toward deciphering the complex coordination between enzymatic turnover and phase behavior. Future research is required to resolve how the ATP-dependent 'DNA clamp' cycle is spatially coupled to the physical compaction of chromatin, and whether additional co-repressors are selectively partitioned into these dynamic assemblies to maintain epigenetic stability.

# Materials and methods
## Cell lines
HEK293F cells (Thermo Fisher Scientific, cat. no. R79007) and HEK293T cells (ATCC, cat. no. CRL-11268) were used for protein expression and transfection experiments. HeLa cells (ATCC, cat. no. CRM-CCL-2) were used for transfection, gene editing, and RNA-seq experiments. All mammalian cell lines were sourced directly from certified and qualified suppliers (Thermo Fisher Scientific or ATCC), whose repositories perform routine short tandem repeat (STR) profiling for identity authentication and mycoplasma testing prior to distribution. All cell lines tested negative for mycoplasma contamination as guaranteed by the supplier at the time of purchase, and cultures were periodically verified to be mycoplasma-free during the course of this study. For bacterial strains, *E. coli BL21(DE3*; New England Biolabs, cat. no. C2527H) and *E. coli* B834 (Sigma-Aldrich, cat. no. 69041) were used for recombinant protein expression. Both strains were obtained from certified commercial suppliers and are defined, sequence-verified laboratory strains that do not require STR-based authentication.

## Plasmid construction, cell culture, and transfection

WT *MORC2* (UniProt: Q9Y6X9) cDNA was amplified from a cDNA library derived from HeLa cells via reverse transcription. The *MORC2* cDNA was then subcloned into the p3xFLAG-Myc-CMV-24 vector between BglII and KpnI restriction sites using seamless cloning. The seamless cloning process was facilitated by the ClonExpress II One-Step Cloning Kit (Vazyme, C112-02) according to the manufacturer's instructions. An *EGFP* sequence was subsequently inserted into the BglII site at the N-terminal of *MORC2* to generate the *EGFP-MORC2* construct. To generate deletion mutant constructs of *MORC2* for intracellular studies, specific regions of *MORC2* were targeted and deleted using a polymerase chain reaction (PCR)-based approach. An 18- or 21 bp homologous sequence originating from one of the forward or reverse primers was introduced into the other primer, and the resulting PCR products were subjected to homologous recombination cloning, facilitated by the 18 bp homologous sequence. For protein expression and purification, MORC2FL and truncated MORC2 were cloned into prokaryotic expression vectors with MBP or Trx fusion proteins for bacterial expression.

In addition, MORC2FL and a series of truncation mutants were purified using a HEK293F (Thermo Fisher, R79007) eukaryotic expression system for in vitro phase separation and SLS assays. A260/280 measurements, agarose gel analysis, or EMSA information on protein purifications and their experimental applications in this study is provided in *Supplementary file 2*. HeLa (ATCC, CRM-CCL-2) and HEK293T (ATCC, CRL-11268) cells were cultured in DMEM medium (Biological Industries) and were maintained in a humidified incubator at 37 °C with 5% $CO_2$. Cell transfection was performed using Lipofectamine 3000 (Invitrogen) according to the manufacturer's instructions. Briefly, cells were seeded in appropriate culture plates the day before transfection and allowed to reach 40–60% confluency. The separately diluted Lipofectamine 3000 reagent and plasmids in Opti-MEM reduced serum medium (Gibco) were mixed, and the transfection complexes were added to cells after 20 min. After a 6-hr incubation, the transfection medium was replaced with fresh complete DMEM medium.

## Recombinant protein expression and purification

Plasmids containing His-tagged genes were transformed into *E. coli* BL21 (DE3) cells (NEB, C2527H) and cultured in Luria-Bertani medium with 50 mg/L ampicillin. When the culture reached an optical density (OD) of 0.6–0.8 at 600 nm, the temperature was lowered to 16 °C. Isopropyl β-D-1-thiogalactopyranoside (IPTG; 0.25 mM) was added to induce protein expression for 18–20 hr. Cells were harvested by centrifugation at 13,000 × *g* for 15 min, then resuspended in Buffer A (50 mM Tris-HCl, pH 8.0, 1000 mM NaCl, 10 mM imidazole, 1 mM dithiothreitol [DTT], and 1 mM phenylmethylsulfonyl fluoride [PMSF]). The cell suspension was snap-frozen and stored at –80 °C for subsequent protein purification. HEK293F suspension-adapted cells were cultured in conical flasks at 37 °C, 120 rpm, and 5% $CO_2$, until a density of $1.0\times10^6$ cells/mL was reached. The cells were transiently transfected with 1 mg of *MORC2–6×His* plasmid using 2 mL of branched polyethyleneimine (1 mg/mL). After 48 hr of incubation, cells were harvested by centrifugation at 3000 × *g* for 5 min, resuspended in Buffer A, snap-frozen, and stored at –80 °C for further purification.

All purification steps were performed at 4 °C to preserve protein activity. Frozen cell suspensions were rapidly thawed in water, lysed by sonication (200 W for ~0.3 hr), and centrifuged at 20,000 × *g* for 30 min at 4 °C. The supernatant was filtered through a 0.22 μm syringe filter to obtain soluble material. A His-Trap Ni-NTA 6FF chromatography column was equilibrated with five column volumes of Buffer A (Buffer A: 50 mM Tris-HCl, pH 8.0, 1000 mM NaCl). The clarified extract was loaded onto the column, washed with Buffer A to remove non-specific proteins, and eluted with a linear gradient of nickel elution buffer (Buffer B: 50 mM Tris-HCl, pH 8.0, 1 M NaCl, 300 mM imidazole, 1 mM DTT). Eluted fractions were analyzed by SDS-PAGE and Coomassie staining. The protein of interest was concentrated, exchanged into Buffer C (50 mM Tris-HCl, pH 8.0, 1000 mM NaCl, 1 mM EDTA, 1 mM DTT) using an Amicon Ultra centrifugal filter unit, following overnight treatment with Benzonase nuclease to eliminate nucleic acid contamination or HRV 3 C Protease to cut His-tag. The purification process was subsequently refined through gel filtration using a Superdex 6 Increase column (GE Healthcare) for the MORC2FL. Following this, fragments were conjugated with a GPGS linker, and the resulting mixture was subjected to a size-exclusion chromatography process (Superdex 200/75 column, GE Healthcare) to yield a final solution comprising 50 mM Tris-HCl, pH 8.0, 500 mM NaCl, 1 mM EDTA, and 1 mM DTT. Peak fractions were analyzed by SDS-PAGE and Coomassie staining, confirming the successful purification of highly homogeneous MORC2FL (1–1032) and

various fragments (*Figure 7—figure supplement 2*), then flash-frozen in liquid nitrogen and stored at –80 °C.

## Crystallization, data collection, and processing

The CC3 domain (901–1003) and various mutations of MORC2 constructs were cloned into a modified pET32m vector (with Trx-tag) using standard PCR-based methods and confirmed by DNA sequencing. Due to the lack of homologous structures, protein crystals with heavy atom derivatives are required to determine their phases. Therefore, we used the methionine-deficient strain B834 to express the target protein with selenomethionine (SeMet) in an inorganic selenosubstituted medium. Recombinant proteins were expressed in *E. coli* B834 (Sigma, 69041) cells in M9 medium at 16 °C. MORC2 CC3 fused with a GPGS linker were finally changed by step of size-exclusion chromatography (Superdex 75 increase column, GE Healthcare) into 50 mM Tris-HCl, pH 8.0, 100 mM NaCl, 1 mM EDTA and 1 mM DTT. The freshly purified protein was concentrated to 10~20 mg/mL. Crystals were obtained by the static sitting drop method in 0.1 M Citric acid (pH 3.5) and 25% w/v PEG 3350 at 16 °C. Glycerol (20%) was added as the cryo-protectant. A 3.1 Å resolution X-ray dataset was collected at the beamline BL18U1 of the Shanghai Synchrotron Radiation Facility (SSRF). The diffraction data were processed and scaled by HKL2000 (http://www.hkl-xray.com/). Using the structure of auto-build model by SAD data and AlpahFold2 predicted model as the search model, the initial structural model was solved by molecular replacement in PHASER (https://www.ccp4.ac.uk/). Further manual model adjustment and refinement were completed iteratively using COOT and PHENIX. The final structure was validated by PISA and MolProbity (https://www.phenix-online.org/). The final refinement statistics of the complex structure are listed in *Supplementary file 1*. All structure figures were prepared using the programme PyMOL (http://pymol.sourceforge.net/).

## Protein labeling with fluorophore

The Cy3 NHS ester (AAT Bioquest) fluorophores were dissolved in DMSO at 10 mg/mL. Purified proteins were prepared in reaction buffer at a concentration of 10 mg/mL. The fluorophores were added to the protein solution at a 1:1 molar ratio. The reaction was conducted with shaking at room temperature for 1 hr in the dark. After the reaction, labeling was quenched with 200 mM Tris-HCl, pH 8.3, and the labeled proteins were desalted using a HiTrap desalting column (GE Healthcare). Fluorescence labeling efficiency was assessed using a Nanodrop 2000 spectrophotometer (Thermo Fisher, ND-2000). The final labeling efficiency of each protein was adjusted to 1% by mixing the labeled protein with an excess of unlabeled protein (1:99 ratio).

## Lentiviral mediated H2A-mCherry inducible expression in HeLa cells

Stable expression of H2A-mCherry in HeLa cells was achieved through lentiviral-mediated transduction. First, *H2A-mCherry* gene sequence was inserted into the inducible lentiviral vector pTRIPZ to generate the expression plasmid pTRIPZ-H2A-mCherry. Lentiviral particles were generated by triple transfection of HEK293T cells with pTRIPZ-H2A-mCherry, PMD2.G, and PSPAX2 plasmids using Lipofectamine 3000. The viral supernatant was collected 48 hr after transfection, filtered with 0.45 µm filter to remove cell debris, and used to transduce HeLa cells. Following transduction, HeLa cells were selected with 50 µg/mL hygromycin to establish stable cell lines. H2A-mCherry expression was induced with 1 µg/mL doxycycline.

## Live cell imaging and immunofluorescence staining

For live-cell imaging, HeLa cells were seeded onto 35 mm culture dishes with glass-bottom plates (Biosharp) and transfected with *EGFP-MORC2* or its mutant constructs. Images were captured 12 hr post-transfection using a Nikon confocal microscope equipped with an incubator system (37 °C, 5% $CO_2$). Nuclei were stained with Hoechst 33342 or visualized using mCherry-labeled H2A fusion protein. For immunofluorescence, cells were fixed with 4% paraformaldehyde, permeabilized with 0.1% Triton X-100, and blocked with 10% goat serum. Primary antibodies were incubated overnight at 4 °C, followed by incubation with fluorescence-conjugated secondary antibodies for 2 hr at room temperature in the dark. Fluorescence images were acquired using a Nikon confocal microscope.

## CRISPR-Cas9-mediated *MORC2*-knockout cells

*MORC2*-knockout (KO) HeLa cell lines were generated using the CRISPR-Cas9 system. A CRISPR single-guide RNA (sgRNA) targeting exon 5 of *MORC2* (5′-ACACCTGAGTCTACTCAGAT-3′) was designed

using the online tool (http://crispor.tefor.net/). Two DNA oligos with complementary sgRNA targeting sequence and additional bases were designed as previously described (*Shalem et al., 2014*; *Sanjana et al., 2014*). These oligos were annealed and cloned into pLentiCRISPR v2 (Addgene, 52961) to generate the *MORC2*-knockout vector. After transfection and puromycin selection, *MORC2*-deleted cells were obtained. Individual cells were sorted into separate wells of a culture plate to establish monoclonal cell lines using flow cytometry. To validate the knockout cell lines, genomic DNA was extracted from both WT and KO cells. The specific regions targeted by the sgRNA were amplified using PCR with primers: 5′-*CAACATTCTCGAGCTGGACCTACAG-3′* (forward) and 5′-*CCACGACA AGACTGGAAACGTGACTC-3′* (reverse). The PCR products were purified and subjected to Sanger sequencing, revealing the deletion and insertion mutations in the target sites of the *MORC2* allelic genes (*Figure 2—figure supplement 1*).

## Generation of endogenous *Morc2a^EGFP^* chimeric mice

CRISPR-Cas9 was used to generate *Morc2a^EGFP^* chimeric mice. The EGFP and linker sequence were inserted at the N-terminus of the mouse *Morc2a* gene. Briefly, a single-guide RNA (sgRNA) was designed to target exon 1 of *Morc2a*, with the sequence *gactgaagactcattgctgtca*. A donor DNA fragment containing the EGFP-linker region flanked by homologous arms was constructed. Fertilized mouse zygotes were collected, and a mixture of Cas9 mRNA (50 ng/μL), sgRNA (25 ng/μL), and donor DNA (50 ng/μL) was microinjected into the pronucleus of the zygotes. The injected embryos were transferred into the oviducts of pseudopregnant recipient female mice. Resulting pups were genotyped by PCR and Sanger sequencing to confirm transgene integration (*Figure 2—figure supplement 2*). The genotyping primers for the *Morc2a^EGFP^* allele were as follows: Left-F1: *gcggtggttgagttcc aattcc*; Left-R1: *cttcagggtcagcttgccgtag*; Right-F1: *gaacggcatcaaggtgaacttc*; Right-R1: *tgtatccaaggt aatctctgtggta*.

Mice were bred at the Animal Center of the University of Science and Technology of China (USTC). Animals were housed under controlled conditions (22 °C, 12 hr light/dark cycle) with free access to food and water ad libitum. All experimental procedures involving animals were approved by the Institutional Animal Care and Use Committee (IACUC) of the University of Science and Technology of China (Hefei, China), under protocol number PXHG-WYC2021072713.

## Fluorescence recovery after photobleaching (FRAP)

The FRAP assay was performed to investigate the dynamics of protein mobility within condensates, either in in vivo or in vitro. Specific regions of interest (ROIs) approximately 2 μm² in size were selected and subjected to photobleaching using 100% laser power (488 nm) for approximately 20 ns. One droplet was left unbleached as a control. Fluorescence recovery within the ROI was monitored for 100–300 s. FRAP was performed using a 63 x oil immersion objective mounted on a Nikon confocal microscope or a Zeiss LSM 880. Fluorescence intensity was normalized to the pre-bleach signal to analyze the recovery kinetics. For brain slice FRAP, organotypic brain slices were prepared from adult chimeric mice expressing endogenous EGFP-MORC2, following a slightly modified protocol previously described (*Li et al., 2020*). After decapitation, brains were rapidly extracted and placed in ice-cold dissection medium consisting of Hibernate A, 2% B27 supplement, 2 mM L-glutamine, and 1% penicillin–streptomycin. The cerebellum and midbrain were removed, and the remaining cerebral hemispheres were coronally sectioned at 250 μm using a McIlwain tissue chopper. Slices were gently separated in chilled dissection medium and transferred onto glass-bottom dishes containing culture medium composed of Neurobasal A, 2% B27 supplement, 2 mM L-glutamine, and 1% penicillin–streptomycin. Imaging was initiated immediately after brain slice preparation. FRAP experiments were performed using the Nikon confocal microscope. Photobleaching was carried out with 5–7 laser pulses (50 μs dwell time), and images were acquired at 2 s intervals. Post-bleach images were acquired 20 s after photobleaching.

## Sedimentation assay

To assess concentration-dependent partitioning behavior, purified proteins at the indicated concentrations were incubated under the same buffer conditions used for in vitro phase separation assays and equilibrated for 10 min at room temperature. Samples were then centrifuged at 16,000 × *g* for 10 min (25 °C) to separate the supernatant and pellet fractions. Following centrifugation, the supernatant was

carefully collected, and the pellet was resuspended in an equal volume of buffer. Equal volumes of supernatant and pellet fractions were analyzed by SDS–PAGE and quantified by densitometry.

## Electrophoretic mobility shift assays (EMSA)

Widom 601 DNA (147 bp, 25 nM) was amplified by PCR and incubated with increasing concentrations of MORC2 or its domain constructs in binding buffer (20 mM Tris-HCl, pH 7.5, 45 mM KCl, 4 mM MgCl$_2$, 5% glycerol, 5 mM DTT). The reaction mixture was loaded onto a 6% polyacrylamide gel and electrophoresed at 120 V for 70 min. DNA was visualized using SYBR Gold stain, and images were captured using the ChemiDoc MP Imaging System.

## Size-exclusion chromatography coupled with static light scattering

The analysis was performed using an AKTA FPLC system (GE Healthcare) coupled to a SLS detector (miniDawn, Wyatt) and a differential refractive index detector (Optilab, Wyatt). Protein samples (70 μM) were filtered and applied to a Superdex 200 increase or Superose 6 increase column, pre-equilibrated with a column buffer containing 50 mM Tris-HCl, pH 8.0, 150 mM NaCl or 1000 mM NaCl (for MORC2FL), 1 mM EDTA, and 1 mM DTT. Data were analyzed using ASTRA6 software (Wyatt).

## In vitro phase separation and fluorescence microscopy imaging

Droplet formation was monitored using differential interference contrast (DIC) and fluorescence microscopy. To induce phase separation, protein was added to phase separation buffer (50 mM Tris-HCl, pH 8.0, 150 mM NaCl, 1 mM DTT). For consistency, all MORC2 samples were diluted to a final concentration of 4 μM with phase separation buffer containing 200 mM NaCl, unless otherwise specified. All preparation steps were performed at 4 °C to preserve protein activity. The phase separation mixtures were then equilibrated at room temperature for 20 min. A 5–10 μL aliquot of each sample was placed onto a custom-made glass slide, covered with an 18 mm diameter coverslip, and incubated at room temperature for 20 min before imaging.

## Protein isotope labeling and NMR spectroscopy

Isotopically labeled protein samples were expressed in minimal M9/H$_2$O or M9/D$_2$O medium. Uniformly $^{15}$N-labeled (or $^{13}$C- or $^{15}$N-labeled) protein samples were prepared in M9/H$_2$O medium supplemented with 1 g/L of $^{15}$NH$_4$Cl and 2 g/L of glucose (or $^{13}$C-glucose). The cells were harvested at an OD$_{600}$ of approximately 1.0. Isotope-labeled precursors and amino acids were purchased from Cambridge Isotope Laboratories (CIL). Isotopically labeled samples were prepared in 50 mM Tris-HCl, pH 8.0, 100 mM NaCl, and 7% D$_2$O. Protein concentrations ranged from 0.5 to 1.0 mM. NMR experiments were performed using Bruker Avance III 600 MHz spectrometers equipped with a cryo-probe. NMR spectra of MORC2 and its complex were recorded at 298 K. Backbone assignments of MORC2$^{1004-1032}$ were obtained using SOFAST versions of standard triple-resonance experiments, including HNCACB and CBCACONH, recorded at 298 K. Residual assignment ambiguities were resolved using $^{15}$N-edited HMQC-NOESY-HMQC spectra. All NMR spectra were processed using NMRPipe and analyzed with NMRViewJ software (http://www.onemoonscientific.com). CSPs were calculated in Hz at 600 MHz using the following equation:

$$\Delta v = \left( \left( \Delta^1 H \times 600 \right)^2 + \left( \Delta^{13} C \times 150 \right)^2 \right)^{1/2}$$

## Fluorescence polarization assay (FP)

Fluorescence polarization (FP) assays were performed at room temperature (25 °C) using a BioTek H1 fluorescence spectrophotometer to evaluate the binding affinity between MORC2 protein and 5-carboxyfluorescein (5-FAM)-labeled 601 DNA (FAM-601) across a concentration range of 0–2000 nM. The 147 bp Widom 601 DNA fragment was generated via PCR amplification using a 5-FAM-labeled primer. The labeled primer was dissolved in 0.1 M NaHCO$_3$ buffer, pH 8.3, and the amplified product was purified by agarose gel extraction to eliminate residual primers and nonspecific amplification products.

Fluorescence measurements were carried out in assay buffer (50 mM Tris-HCl, pH 8.0, 150 mM NaCl, 1 mM EDTA, 1 mM DTT, and 10% glycerol) containing purified FAM-601 DNA at a final

concentration of 0.4 ng/µL. Excitation and emission wavelengths were optimized according to the instrument manual, with excitation set at 485 nm and emission at 525 nm. Fluorescence polarization values were recorded in triplicate, and control measurements were included to correct for background signals.

Data were analyzed to determine the equilibrium dissociation constant ($K_d$) for the MORC2-FAM-601 interaction by fitting data to a log(agonist) vs. response model. In parallel, the effect of small molecules on this binding interaction was assessed by introducing compounds into the assay mixture and monitoring changes in fluorescence polarization.

## ATPase activity measurements

ATPase activity was measured (*Gawronski and Benson, 2004*) using 1 µM MORC2 or its mutant variants at 37 °C in 50 mM Tris-HCl, pH 8.0, 150 mM NaCl, 8 mM MgCl$_2$, 10% glycerol, and 5 mM ATP. All experiments were repeated independently at least three times.

## RNA-seq and bioinformatic analysis

*MORC2*-knockout HeLa cells were seeded at equal density and transfected with 3 µg of plasmid DNA and 12 µg of PEI (polyethylenimine) per dish. After 6 hr, the transfection medium was replaced with fresh complete medium, and cells were incubated for an additional 48 hr. Cells were then washed twice with ice-cold PBS and lysed directly in RNA Isolator (R401, Vazyme) on ice. Total RNA was extracted using the RNeasy Mini Kit (Qiagen, 74004) according to the manufacturer's protocol. RNA quality was assessed using NanoDrop 2000 (Thermo Fisher Scientific), Agilent 2100 Bioanalyzer, and LabChip GX (PerkinElmer). For library preparation, 1 µg of total RNA was used per sample, employing the VAHTS Universal V6 RNA-seq Library Prep Kit for MGI. The resulting double-stranded DNA libraries were denatured, circularized, and digested to obtain single-stranded circular DNA. DNA nanoballs (DNBs) were generated from these templates through rolling circle amplification (RCA). Sequencing was performed on the MGI DNBSEQ-T7 platform (MGI Tech Co., Ltd) using combinatorial Probe-Anchor Synthesis (cPAS) chemistry.

Before performing differential expression analysis, lowly expressed genes were filtered out by retaining only those with a raw count of ≥10 in at least three samples (baseMean). Differential gene expression analysis between sample groups was carried out using the DESeq2 package (v1.26.0). Genes with *p-adjust*<0.05 were defined as differentially expressed genes (DEGs). These DEGs were subsequently subjected to Gene Ontology (GO) enrichment analysis in DAVID (Knowledgebase v2025_1). The raw gene expression data are provided in *Supplementary file 3*.

## Statistical analysis

All statistical analyses were performed in GraphPad Prism. All data underwent normality testing (D'Agostino and Pearson test and Anderson-Darling test) in advance. Two-tailed unpaired Student's *t*-test post-hoc Welch's correction and one-way ANOVA with Tukey's post hoc test was used when data were distributed normally. The significance level was set as $p < 0.05$. All statistical results are provided as Source Data.

## Acknowledgements

We thank Dr. Xuebiao Yao (University of Science and Technology of China, USTC), Dr. Zhi Qi and Dr. Linyu Zuo (Peking University) for helpful discussion. We thank members of the Shanghai Synchrotron Radiation Facility (SSRF, China; beamlines BL02U1, BL18U1, and BL19U1) for X-ray beam time. This work was supported by grants from the National Key R&D Program of China (2019YFA0508402), the National Natural Science Foundation of China (22122703, 32470808, 32170767, 32471273, 31971144, and T2221005), the Center for Advanced Interdisciplinary Science and Biomedicine of IHM (QYPY20220014), the Strategic Priority Research Program of the Chinese Academy of Sciences (XDB0490000), the Major Frontier Research Project of USTC (LS9100000002), USTC Research Funds of Double First-Class Initiative (YD9100002507), the Anhui Medical University Foundation (9101224201), and National Natural Science Foundation Incubation Program of The Second Affiliated Hospital of Anhui Medical University (2021GQFY02).

## Additional information

### Funding

| Funder | Grant reference number | Author |
|---|---|---|
| National Natural Science Foundation of China | 22122703 | Chao Wang |
| National Key Research and Development Program of China | 2019YFA0508402 | Chao Wang |
| National Natural Science Foundation Incubation Program of The Second Affiliated Hospital of Anhui Medical University | 2021GQFY02 | Yihui Bi |
| National Natural Science Foundation of China | 32470808 | Chao Wang |
| National Natural Science Foundation of China | 32170767 | Chao Wang |
| National Natural Science Foundation of China | 32471273 | Chengdong Huang |
| National Natural Science Foundation of China | 31971144 | Chengdong Huang |
| National Natural Science Foundation of China | T2221005 | Chengdong Huang |
| Center for Advanced Interdisciplinary Science and Biomedicine of IHM | QYPY20220014 | Chao Wang |
| Strategic Priority Research Program of the Chinese Academy of Sciences | XDB0490000 | Chao Wang |
| Major Frontier Research Project of USTC | LS9100000002 | Chao Wang |
| USTC Research Funds of Double First-Class Initiative | YD9100002507 | Chao Wang |
| Anhui Medical University Foundation | 9101224201 | Yihui Bi |

The funders had no role in study design, data collection and interpretation, or the decision to submit the work for publication.

### Author contributions

Yanshen Zhang, Conceptualization, Resources, Data curation, Software, Formal analysis, Validation, Investigation, Visualization, Methodology, Writing – original draft, Writing – review and editing; Weiya Xu, Data curation, Software, Formal analysis, Validation, Investigation, Visualization, Methodology; Wenxiu Duan, Conceptualization, Resources, Data curation, Software, Formal analysis, Validation, Investigation, Visualization, Methodology, Writing – original draft; Yu Wei, Resources, Data curation, Formal analysis, Investigation, Methodology; Wenli Jiang, Resources, Data curation, Investigation; Feng Zhu, Resources, Methodology; Chengdong Huang, Conceptualization, Resources, Data curation, Software, Formal analysis, Supervision, Funding acquisition, Validation, Investigation, Visualization, Methodology, Project administration, Writing – review and editing; Chao Wang, Conceptualization, Resources, Formal analysis, Supervision, Funding acquisition, Writing – original draft, Project administration, Writing – review and editing; Yihui Bi, Conceptualization, Resources, Data curation, Software, Formal analysis, Funding acquisition, Validation, Investigation, Visualization, Methodology, Writing – original draft, Writing – review and editing

## Author ORCIDs

Yanshen Zhang ⓘ https://orcid.org/0000-0001-8046-0389
Weiya Xu ⓘ https://orcid.org/0000-0001-5796-8758
Chengdong Huang ⓘ https://orcid.org/0000-0002-8997-9459
Chao Wang ⓘ https://orcid.org/0000-0003-3192-2780
Yihui Bi ⓘ https://orcid.org/0009-0000-5486-9009

## Ethics

Mice were bred at the Animal Center of the University of Science and Technology of China (USTC). Animals were housed under controlled conditions (22°C, 12-hr light/dark cycle) with free access to food and water ad libitum. All experimental procedures were approved by the Institutional Animal Care and Use Committee (IACUC) of the University of Science and Technology of China (Hefei, China) under protocol number PXHG-WYC2021072713.

Reviewer #1 (Public review): https://doi.org/10.7554/eLife.108479.3.sa1
Reviewer #2 (Public review): https://doi.org/10.7554/eLife.108479.3.sa2
Reviewer #3 (Public review): https://doi.org/10.7554/eLife.108479.3.sa3
Author response https://doi.org/10.7554/eLife.108479.3.sa4

# Additional files

## Supplementary files

Supplementary file 1. Statistics of X-ray Crystallographic Data Collection and Model refinement Of MORC2_CC3 Domain.

Supplementary file 2. Summary of MORC2 protein constructs used in this study, including expression systems, extinction coefficients, A260/280 ratios, and corresponding experimental applications.

Supplementary file 3. The raw gene expression data for *Figure 6d*.

MDAR checklist

## Data availability

All data needed to evaluate the conclusions in the paper are present in the paper and/or the supplementary materials. Raw gene expression data are available as source data. The atomic coordinates of the MORC2 CC3 domain have been deposited to the Protein Data Bank under the accession code: 9KQL.

The following dataset was generated:

| Author(s) | Year | Dataset title | Dataset URL | Database and Identifier |
|---|---|---|---|---|
| Zhang YS, Xu WY, Huang CD, Wang C | 2025 | The crystal structure of MORC2_CC3 domain at 3.1 Angstroms resolution | https://doi.org/10.2210/pdb9kql/pdb | Worldwide Protein Data Bank, 10.2210/pdb9kql/pdb |

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
