## [Editor Report · eLife Assessment]

This **important** study has demonstrated that MORC2 undergoes phase separation in cells and established multiple interactions responsible for the phase separation. Upon revision, the data generally provide **solid** support to the claim that MORC2 condensates are functionally relevant in gene regulation and begins to demonstrate the importance of the physical properties of biological condensates. Nevertheless, there remains some weakness in the connection between condensates and function.

---

## [Referee Report · Reviewer #1 (Public review)]

This work demonstrates that MORC2 undergoes phase separation (PS) in cells to form nuclear condensates, and the authors demonstrate convincingly the interactions responsible for this phase separation. Specifically, the authors make good use of crystallography and NMR to identify multiple protein:protein interactions and use EMSA to confirm protein:DNA interactions. These interactions work together to promote in vitro and in cell phase separation and boosted ATPase activity by the catalytic domain of MORC2.

Moreover, the authors show solid evidence supporting their important claim that MORC2 PS is important for MORC2-mediated gene regulation. Exploring causal links between PS and function is an important need in the phase separation field, particularly as regards the role of condensates in gene regulation, and is a non-trivial matter. It is crucial and challenging to properly explore the alternative possibility that soluble complexes, existing in the same conditions as phase-separated condensates, are the functional species. The authors have attempted to address this concern by manipulating the physical nature of the MORC2 condensates using a killswitch (KS) peptide (MORC2 +KS), finding that reducing condensates dynamics results in a cellular phenotype very similar to that of the phase separation-deficient MORC2 condensates. While not fully ruling out the alternative, soluble-complex hypothesis, this experiment suggests that function is indeed localized inside the MORC2 condensates, and that perturbing the condensate can be functionally equivalent to removing condensate formation.

The authors also look at several disease related mutants of MORC2. While most of these do not seem to have an obvious connection to the phase separation data, it is quite interesting that one mutant, E236G, displays similar intra-condensate dynamics compared to MORC2 +KS, strengthening the claim that MORC2 phase separation is important for function and suggesting that the observations in this paper may indeed have some disease relevance.

Strengths

Static light scattering and crystallography are nicely used to demonstrate the dimerization of MORC2FL and to discover the structure of the CC3 domain dimer, presumably responsible for the dimerization of MORC2FL (Figure 1).

Extensive use of deletion mutants in multiple cell lines is used to identify regions of MORC2 that are important for forming condensates in the nucleus: the IBD, IDR, and CC3 domains are found to both be essential for condensate formation, while the CW domain plays an unknown role in condensate morphology (Figure 3). The authors use NMR to further identify that the IBD domain seems to interact with the first third of the centrally located IDR, termed IDRa, but not with the latter two thirds of the IDR domain (Figure 4). This leads them to propose that phase separation is the product of IDB:IDRa interaction, CC3 dimerization, and an unknown but important role for the CW domain.

Based on the observation that removal of the NLS resulted in diffuse cytoplasmic localization, they hypothesized that DNA may play an important role in MORC2 PS. EMSA was used to demonstrate interaction between DNA and several MORC2 domains: CC1, CC2, IDR, and TCD-CC3-IBD. Further in vitro microscopy with purified MORC2 showed that DNA addition significantly reduces MORC2 saturation concentration (Figure 5).

These assays convincingly demonstrate that MORC2 phase separates in cells and identifies the protein domains and interactions responsible for this phenomenon.

Weaknesses

The connection between condensates and function, while improved from the original manuscript, still has some weak points.

The central experiment demonstrating that MORC2 condensates mediate function takes the form of RNA-Seq in MORC2 KO HeLa cells (Figure 6), rescued with WT, condensate-deficient mutants, and a KS peptide mutant that reduces dynamics by increasing homotypic protein interactions. The observation that rescuing with MORC2 +KS is ineffective, in a manner similar to rescue with condensate-deficient MORC2 mutants, suggests that unperturbed condensates are important for function. An alternative possibility, however, is that condensates are non-functional bystanders, and that the increased homotypic interactions present in MORC2 +KS result in stronger MORC2 +KS recruitment to condensates, reducing the pool of functional, dilute phase MORC2 +KS and squashing function via sequestration. Similar ideas have been explored by others for transcription factors (e.g. Chong et al, Mol Cell, 2022). This possibility is neither discussed nor ruled out. The absence of microscopy data showing similar localization of MORC2 and MORC2 +KS (particularly the amount of diffuse MORC2 outside condensates) amplifies this concern.

The RNA-Seq data presented in Figure 6h also has some concerning qualities. Inter-replicate variability is higher than ideal, particularly for MORC2 deltaCC3. This may be a product of the transient transfection system used for these experiments, which inherently results in stochasticity. Specific sets of genes regulated by MORC2 are consistent with the main conclusion (Figure 6i, individual genes in 6h, showing that all mutants are more similar to one another than to WT MORC2), but global transcription shifts seem quite different between MORC2 condensate-deficient mutants and MORC2 +KS (Figure 6h heatmap), suggesting much more than simple condensate disruption is taking place. Together, this weakens the conclusion that MORC2 condensates are the functional form of MORC2.

---

## [Referee Report · Reviewer #2 (Public review)]

Summary:

The study by Zhang et al. focuses on how condensation of a chromatin-associated protein MORC2 regulates gene expression. Their study shows that MORC2 forms dynamic nuclear condensates in cells. In vitro, MORC2 phase separation is driven by dimerization and multivalent interactions involving the C-terminal domain but interplay with other parts of MORC2 too. A key finding is that the intrinsically disordered region (IDR) of MORC2 exhibits strong DNA binding. They report that DNA binding enhances MORC2's phase separation and its ATPase activity, offering new insights into how MORC2 contributes to chromatin organization and gene regulation. Authors correlate MORC2's condensate forming ability and material properties with its gene silencing function using a few variants. Moreover, they investigate the effect of disease-linked mutations in the N-terminal domain of MORC2 on its ability to form cellular condensates, ATPase activity and DNA-binding. Their work implies that proper material properties of MORC2 condensates may be important to their biological function.

Strengths:

The authors determined a 3.1 Å resolution crystal structure of the dimeric coiled-coil 3 (CC3) domain of MORC2, revealing a hydrophobic interface that stabilizes dimer formation. They present extensive evidence that MORC2 phase separates across multiple contexts, including in vitro, in cellulo, and in vivo. Through systematic cellular screening, they identified the C-terminal domain of MORC2 as a key driver of condensate formation. Biophysical and biochemical analyses further show that the IDR within the C-terminal domain interacts with the C-terminal end region (IBD) and also exhibit strong DNA-binding capacity (using 601 DNA), both of which promote MORC2 phase separation. Together, this study emphasizes that interactions mediated by multiple domains-CC3, IDR, and IBD- drives MORC2 phase separation. Additionally, the work uses a unique kill-switch peptide fused to the MORC2 sequence to disrupt its material properties -- this permits the authors to examine the link between material properties and transcription function. The study is overall strengthened by (1) the combination of variants tested both in vitro and in cellulo, and (2) the systematic examination of domain contributions that highlight the multivalent interactions at play mediating MORC2 condensation.

Weaknesses:

The employed MORC2 variants have enabled the beginning of an investigation linking condensation and biological function, but more work will be needed to really dissect the contribution of condensation to DNA-binding, ATPase activity, and gene silencing. A systematic investigation of differential material properties on MORC2 condensates will be needed to assess the link to biological function, especially as the authors' work is reminiscent of how the liquidity of *Caulobacter crescentus* PopZ condensates tunes bacterial fitness.

---

## [Referee Report · Reviewer #3 (Public review)]

Summary:

The manuscript by Zhang et al. demonstrates that MORC2 undergoes liquid-liquid phase separation (LLPS) to form nuclear condensates critical for transcriptional repression. Using a combination of in vitro LLPS assays, cellular studies, NMR spectroscopy, and crystallography, the authors show that a dimeric scaffold formed by CC3 drives phase separation, while multivalent interactions between an intrinsically disordered region (IDR) and a newly defined IDR-binding domain (IBD) further promote condensate formation. Notably, LLPS enhances MORC2 ATPase activity in a DNA-dependent manner and contributes to transcriptional regulation, establishing a functional link between phase separation, DNA binding, and transcriptional control.

Strengths:

The manuscript is well organized and logically structured. It provides valuable mechanistic insights into MORC2 function, and the majority of the conclusions are well supported by the data presented.

---

## [Author Response]

The following is the authors’ response to the original reviews.

**Reviewer #1 (Public review):**
Summary:This work demonstrates that MORC2 undergoes phase separation (PS) in cells to form nuclear condensates, and the authors demonstrate convincingly the interactions responsible for this phase separation. Specifically, the authors make good use of crystallography and NMR to identify multiple protein: protein interactions and use EMSA to confirm protein: DNA interactions. These interactions work together to promote in vitro and in cell phase separation and boost ATPase activity by the catalytic domain of MORC2.However, the authors have very weak evidence supporting their potentially valuable claim that MORC2 PS is important for the appropriate gene regulatory role of MORC2 in cells. Exploring causal links between PS and function is an important need in the phase separation field, particularly as regards the role of condensates in gene regulation, and is a non-trivial matter. Any study with convincing data on this matter will be very important. For this reason, it is crucial to properly explore the alternative possibility that soluble complexes, existing in the same conditions as phase-separated condensates, are the functional species. It is also critical to keep in mind that, while a specific protein domain may be essential for PS, this does not mean its only important function pertains to PS.In this study, the authors do not sufficiently explore the role that soluble MORC2 complexes may play alongside MORC2 condensates. Neither do they include enough data to solidly show that domain deletion leads to phenotypes via a loss of phase separation per se, rather than the loss of phase separation being a microscopically visible result, not cause, of an underlying shift in protein function. For these reasons, the authors' conclusions regarding the functional role of MORC2 condensates are based on incomplete data. This also dampens the utility of this work as a whole, since the very nice work detailing the mechanism of MORC2 PS is not paired with strong data showing the importance of this observation.

We thank the reviewer for this thoughtful and constructive critique. We agree that establishing a causal link between phase separation (PS) and biological function—particularly in transcriptional regulation—is a central and non-trivial challenge in the condensate field. We also appreciate the reviewer’s emphasis on two critical alternative interpretations: (i) that soluble MORC2 complexes, rather than condensates, may represent the primary functional species, and (ii) that loss of phase separation upon domain deletion could reflect a downstream consequence of altered protein function rather than its cause.

To address these concerns, we have performed a series of new experiments specifically designed to decouple condensate formation, and condensate dynamics, thereby allowing us to more rigorously interrogate the functional relevance of MORC2 condensates.

First, to overcome the limitation of domain deletions which may affect MORC2 function beyond phase separation we introduced a micropeptide-based kill switch (KS) to the C terminus of MORC2. This strategy has recently emerged as a powerful approach to selectively reduce condensate dynamics without disrupting protein expression, folding, or domain architecture [1]. Importantly, unlike CC3 or IDRa deletions, MORC2+KS robustly form nuclear condensates but exhibits markedly reduced internal dynamics, as demonstrated by FRAP analyses showing minimal fluorescence recovery after photo bleaching (Fig. 6a-c). This strategy therefore allows us to perturb condensate material properties independently of MORC2 domain integrity.

Second, we systematically compared the transcriptional consequences of rescuing MORC2-knockout HeLa cells with MORC2FL, condensation-deficient mutants (ΔCC3 and ΔIDRa), and the dynamics-defective MORC2+KS (Fig. 6d). Despite being expressed at substantially higher levels than MORC2FL (Fig. 6e), all three mutants showed a striking and consistent failure to restore MORC2-dependent transcriptional regulation (Fig. 6f-h). This effect was particularly pronounced for transcriptionally repressed genes, including two sets of high-confidence MORC2 targets reported in prior studies (Fig. 6i and Fig.S10). These findings demonstrate that neither increased protein abundance nor the mere presence of condensate-like structures alone is sufficient to restore MORC2 function.

Third, our data instead support a model in which both soluble MORC2 complexes and dynamic MORC2 condensates are required for full transcriptional regulation activity. While soluble MORC2 is likely involved in target recognition and complex assembly, our results indicate that proper condensate formation—and critically, condensate dynamics—are essential for effective transcriptional repression and activation. The inability of the MORC2+KS mutant to rescue transcriptional defects, despite intact condensate formation, points away from a model in which MORC2 condensates represent only microscopically visible byproducts of MORC2 activity.

We believe these new data strengthen the manuscript by pairing the detailed mechanistic dissection of MORC2 phase separation with direct functional evidence, enhancing the conceptual impact and biological significance of the study.

Strengths:Static light scattering and crystallography are nicely used to demonstrate the dimerization of MORC2FL and to discover the structure of the CC3 domain dimer, presumably responsible for the dimerization of MORC2FL (Figure 1).Extensive use of deletion mutants in multiple cell lines is used to identify regions of MORC2 that are important for forming condensates in the nucleus: the IBD, IDR, and CC3 domains are found to be essential for condensate formation, while the CW domain plays an unknown role in condensate morphology (Figure 3). The authors use NMR to further identify that the IBD domain seems to interact with the first third of the centrally located IDR, termed IDRa, but not with the latter two-thirds of the IDR domain (Figure 4). This leads them to propose that phase separation is the product of IDB:IDRa interaction, CC3 dimerization, and an unknown but important role for the CW domain.Based on the observation that removal of the NLS resulted in diffuse cytoplasmic localization, they hypothesized that DNA may play an important role in MORC2 PS. EMSA was used to demonstrate interaction between DNA and several MORC2 domains: CC1, CC2, IDR, and TCD-CC3-IBD. Further in vitro microscopy with purified MORC2 showed that DNA addition significantly reduces MORC2 saturation concentration (Figure 5).These assays convincingly demonstrate that MORC2 phase separates in cells, and identify the protein domains and interactions responsible for this phenomenon, with the notable caveat that the role of the CW domain here is left unexplored.

We appreciate the reviewer for their positive and detailed assessment of the strengths of our study. Our understanding of the CW domain’s function remains preliminary. Although we observed that the CW domain can influence condensate size, the IDR, IBD, and CC3 domains constitute the core structural elements driving phase separation. Consequently, the CW domain was not a primary focus of the current study. Nonetheless, investigating its functional contributions represents an interesting avenue for future work.

Weaknesses:Although the authors demonstrated phase separation of MORC2FL, their evidence that this plays a functional role in the cell is incomplete.Firstly, looking at differentially upregulated genes under MORC2FL overexpression, the authors acknowledge that only 10% are shared with differentially regulated genes identified in other MORC2FL overexpression studies (Figure 6c, d). No explanation is given for why this overlap is so low, making it difficult to trust conclusions from this data set.

We thank the reviewer for raising this important concern. In response, we have improved the quality and robustness of our RNA-seq analysis by repeating the experiments with optimized sample handling and increased sequencing depth. Using this updated dataset, we identified a considerably higher overlap between MORC2-regulated genes in our study and those reported previously.

Specifically, we observed 84 overlapping genes with the study by Nikole L. Fendler et al. [2], corresponding to approximately 32% of the MORC2-regulated genes reported in that work (Fig. 6i). In addition, we identified 102 overlapping genes with the dataset reported by Iva A. Tchasovnikarova et al. [3], representing approximately 22% of the genes identified in that study (Fig. S10b).

We note that complete concordance with previous reports is not expected, given substantial differences in experimental design. For example, Fendler et al. employed a doxycycline-inducible MORC2 expression system [2], whereas our study relies on transient overexpression in MORC2-knockout HeLa cells. In contrast, Tchasovnikarova et al. compared transcriptomes between MORC2 knockout and wild-type cells [3], rather than MORC2 rescue conditions. Moreover, RNA-seq results are inherently influenced by cell line batch variability, sequencing depth, and analysis pipelines, all of which differ across studies.

Taken together, we consider an overlap in the range of ~20–30% to be reasonable and biologically meaningful in the context of these experimental differences, and we believe that the revised RNA-seq data provide a more reliable foundation for our conclusions regarding MORC2-dependent transcriptional regulation.

Secondly, of the 21 genes shared in this study and in earlier studies, the authors note that the differential regulation is less pronounced when a phase-separation-deficient MORC2 mutant is overexpressed, rather than MORC2FL (Figure 6e). This is taken as evidence that phase separation is important for the proper function of MORC2. However, no consideration is made for the alternative possibility that the mutant, lacking the CC3 dimerization domain, may result in non-functional complexes involving MORC2, eliminating the need for a PS-centric conclusion. To take the overexpression data as solid evidence for a functional role of MORC2 PS, the authors would need to test the alternative, soluble complex hypothesis. Furthermore, there seems to be low replicate consistency for the MORC2 mutant condition (Figure S6a), with replicate 3 being markedly upregulated when compared to replicates 1 and 2.

We thank the reviewer for raising these important concerns. In the revised manuscript, we have substantially strengthened both the experimental evidence and the data presentation to directly address the alternative “soluble complex” interpretation as well as the issue of replicate consistency. Specifically, we now provide data that clarify the functional impact of phase-separation-deficient MORC2 mutants and explicitly show replicate-level RNA-seq analyses. The Fig. 6 and Fig. S10support these improvements and enhance both the robustness and transparency of our transcriptional analyses. Collectively, these revisions directly address the reviewer’s concerns regarding the functional interpretation of MORC2 phase separation.

Thirdly, the authors close by examining the in-cell PS capabilities and ATPase activity of several disease-associated mutants of MORC2 (Figure 7). However, the relevance of these mutants to the past 6 figures is unclear. None of these mutations is in regions identified as important for PS. Two of the mutations result in a higher percentage of the cell population being condensate-positive, but this is not seemingly connected to ATPase activity, as only one of these two mutants has increased ATPase activity. Figure 7 does not add any support to the main hypotheses in the paper, and nowhere in the paper do the authors investigate the protein regions where the mutations in Figure 7 are found.

We thank the reviewer for raising this point regarding Fig. 7. At the current stage, the results for disease-associated mutations are primarily descriptive. While we observed that certain mutations clustered at the N-terminus can affect MORC2 condensate formation, ATPase activity, and DNA binding, we did not identify a mechanistic explanation for these correlations. Notably, the T424R mutation, previously reported to significantly enhance ATPase activity [4], also increased both intracellular condensate formation and in vitro DNA binding in our experiments. In contrast, other mutations did not show such consistent effects. Previous studies have established that MORC2’s ATP-binding and DNA-binding activities are independent [4]. Our results further suggest that MORC2’s phase separation behavior is independent of both ATP and DNA binding affinity, although existing evidence hints at potential cross-regulatory interactions among these three functions.

We would also like to emphasize an additional observation that may help contextualize the relevance of N-terminal mutations. Although deletion of the MORC2 N-terminus does not prevent the remaining C-terminal region from forming nuclear condensates, these C-terminal condensates exhibit a marked loss of fluorescence recovery in FRAP assays (Fig. S11). This finding suggests that while the N-terminus is not strictly required for condensate assembly, it plays an important role in regulating condensate fluidity. Accordingly, disease-associated mutations distributed across the N-terminal region may influence MORC2 function by modulating condensate material properties rather than condensate formation per se. Based on this hypothesis, we evaluated the fluidity of condensates formed by the E236G and T424R mutants. FRAP measurements indicated substantially reduced fluorescence recovery in E236G, whereas T424R exerted minimal effects (Fig. 7e, f).

Overall, our interpretation of the results in Fig. 7 is still at a preliminary stage. Nevertheless, the role of the MORC2 N-terminus in modulating condensate fluidity, together with the observed impairment caused by the E236G mutation, appears to be robust, although the underlying mechanism remains to be elucidated. We have incorporated additional discussion on this point and consider it an important direction for future study.

**Reviewer #1 (Recommendations for the authors):**
(1) Why does MORC2 overexpression lead to changes in gene regulation that are so different from past MORC2 overexpression studies? This is unsettling to me.(2) Likewise, why is replicate 3 for the MORC2ΔCC3 variant so different from replicates 1 and 2? Perhaps repeating this experiment would be helpful, both for showing better repeatability and perhaps as regards pulling out a stronger phenotype.

We have repeated the experiments and obtained improved data quality.

(3) A better explanation of the relevance of Figure 7 to the story of the rest of the paper, especially the phase-separation of MORC2, would be important to improving this paper.

We thank the reviewer for this suggestion. We have performed additional experiments and expanded the discussion.

(4) Are expression levels of mutant proteins in Figure 7 uniform between mutants? If not, is it possible that expression levels might account for the difference in condensate-positive cells between mutants?

We cannot fully exclude the possibility that differences in expression levels may contribute to the observed differences among mutants. In our experiments, equal amounts of plasmid DNA were used for transfection across all conditions. Although we did not directly quantify post-transfection protein expression levels by immunoblotting or similar approaches, even if certain mutations were to affect protein expression, it would be technically challenging to further optimize the strategy to fully normalize expression levels across mutants.

Importantly, we note that MORC2 does not form condensates in all transfected cells, even when EGFP fluorescence indicates robust expression levels that are comparable to, or even exceed, those observed in condensate-positive cells. This observation suggests that high expression alone is not sufficient to drive MORC2 phase separation in cells. Therefore, we do not favor the interpretation that the E236K and T424R mutations enhance MORC2 condensation simply by increasing MORC2 protein expression levels.

Minor:(1) I would suggest considering using the term "dynamic" rather than "liquid-like", as FRAP is technically a measurement of the dynamicity of a protein within a volume, rather than a measurement of the actual fluidity of that volume.

We thank the reviewer for this helpful suggestion. We agree that FRAP measurements primarily report protein mobility and condensate dynamics rather than the physical fluidity of the condensates. We have therefore revised the manuscript to replace “liquid-like” with “dynamic” where conclusions are based on FRAP analyses.

(2) A further investigation of the role of the CW domain would be very interesting, since it clearly has a major role in condensate morphology. Perhaps CW confers important heterotypic interactions which contribute to compositional control of the MORC2 condensates, and thus function and morphology? However, due to the complexity of this specific question and the potentially marginal improvement offered by this paper, I do not think this is a critical addition.

We thank the reviewer for this insightful suggestion. We have noted this possibility in the Discussion as an important avenue for future investigation.

(3) Why is TCD not tested alone by EMSA for affinity to DNA in Figure 5?

Our inference regarding the DNA-binding capacity of the TCD domain was based on comparative EMSA analyses. Specifically, we found that the TCD–CC3–IBD fragment was able to bind DNA, whereas the CC3–IBD fragment alone showed no detectable DNA binding. From this comparison, we inferred that the TCD domain is responsible for the observed DNA-binding activity.

Because the TCD domain does not affect MORC2 condensate formation, it was not a central focus of the present study, which primarily aims to elucidate the mechanisms underlying MORC2 phase separation and its functional relevance. For this reason, we did not further test TCD alone by EMSA in Figure 5.

**Reviewer #2 (Public review):**
Summary:The study by Zhang et al. focuses on how phase separation of a chromatin-associated protein MORC2, could regulate gene expression. Their study shows that MORC2 forms dynamic nuclear condensates in cells. In vitro, MORC2 phase separation is driven by dimerization and multivalent interactions involving the C-terminal domain. A key finding is that the intrinsically disordered region (IDR) of MORC2 exhibits strong DNA binding. They report that DNA binding enhances MORC2's phase separation and its ATPase activity, offering new insights into how MORC2 contributes to chromatin organization and gene regulation. The authors try to correlate MORC2's condensate-forming ability with its gene silencing function, but this warrants additional controls and validation. Moreover, they investigate the effect of disease-linked mutations in the N-terminal domain of MORC2 on its ability to form cellular condensates, ATPase activity, and DNA-binding, though the findings appear inconclusive in the manuscript's current form.

Thank you for your thorough and constructive review of our manuscript. In response to the concerns raised regarding the functional relevance of MORC2 condensate formation, we have redesigned and expanded the experiments presented in Fig. 6 and Fig. S6 to directly link MORC2’s condensate-forming capacity with its transcriptional regulatory function. These new experiments provide additional controls and validation, strengthening the causal relationship between MORC2 condensate dynamics and gene regulation.

At the current stage, the results for disease-associated mutations are descriptive. While we observed that certain mutations clustered at the N-terminus can affect MORC2 condensate formation, ATPase activity, and DNA binding, we did not identify a mechanistic explanation for these correlations. Notably, the T424R mutation, previously reported to significantly enhance ATPase activity [4], also increased both intracellular condensate formation and in vitro DNA binding in our experiments. In contrast, other mutations did not show such consistent effects. Previous studies have established that MORC2’s ATP-binding and DNA-binding activities are independent [4]. Our results further suggest that MORC2’s phase separation behavior is also independent of both ATP and DNA binding, although existing evidence hints at potential cross-regulatory interactions among these three functions.

Strengths:The authors determined a 3.1 Å resolution crystal structure of the dimeric coiled-coil 3 (CC3) domain of MORC2, revealing a hydrophobic interface that stabilizes dimer formation. They present extensive evidence that MORC2 undergoes liquid-liquid phase separation (LLPS) across multiple contexts, including in vitro, in cellulo, and in vivo. Through systematic cellular screening, they identified the C-terminal domain of MORC2 as a key driver of condensate formation. Biophysical and biochemical analyses further show that the IDR within the C-terminal domain interacts with the C-terminal end region (IBD) and also exhibits strong DNA-binding capacity, both of which promote MORC2 phase separation. Together, this study emphasizes that interactions mediated by multiple domains-CC3, IDR, and IBD- drives MORC2 phase separation. Finally, the authors quantified the effect of removing the CC3 on the upregulation and downregulation of target gene expression.

We thank the reviewer for their appreciation of the key findings presented in this manuscript.

Weaknesses:Though the findings appear compelling in isolation, the study lacks discussion on how its findings compare with previous studies. Particularly in the context of MORC2-DNA binding, there are previous studies extensively exploring MORC2-DNA binding (Tan, W., Park, J., Venugopal, H. et al. Nat Commun 2025), and its effect on ATPase activity (ref 22). The contradictory results in ref 22 about the impact of DNA-binding on ATPase activity, and ATPase activity on transcriptional repression, warrant proper discussion. The authors performed extensive in-cellulo screening for the investigation of domain contribution in MORC2 condensate formation, but the study does not consider/discuss the possibility of some indirect contributions from the complex cellular environment. Alternatively, the domain-specific contributions could be quantified in vitro by comparing phase diagrams for their variants. While the basis of this study is to investigate the mechanism of MORC2 condensate-mediated gene silencing, the findings in Figure 6 appear incomplete because the CC3 deletion not only affects phase separation of MORC2 but also dimerization. Furthermore, their investigation on disease-linked MORC2 mutations appears very preliminary and inconclusive because there are no obvious trends from the data. Overall, the discussion appears weak as it is missing references to previous studies and, most importantly, how their findings compare to others'.

We thank the reviewer for their careful assessment of MORC2’s DNA-binding properties and its relationship with ATPase and transcriptional activities. We would like to offer the following clarifications to address these concerns, which will also be incorporated into the Discussion section of the revised manuscript.

First, recent work by Tan et al. [5] similarly identified multiple DNA-binding sites in MORC2, consistent with our findings, though there are discrepancies in the precise binding regions. In particular, they reported that isolated CC1 and CC2 domains do not bind 60 bp dsDNA, which contrasts with our observations. We attribute this difference to the types of DNA used in the assays. In our study, we employed 601 DNA, a defined nucleosome-positioning sequence, which differs substantially from randomly designed short dsDNA. For instance, prior work by Christopher H. Douse et al. [54] also confirmed that MORC2’s CC1 domain can bind 601 DNA.

Second, in the study by Fendler et al. [2], DNA binding was reported to reduce MORC2’s ATPase activity—an observation that appears inconsistent with the results presented in our Fig. 5j. A critical distinction between the two studies lies in the experimental systems used: Fendler et al. [2] employed MORC2 constructs and 35 bp double-stranded DNA (dsDNA), whereas our experiments utilized full-length MORC2 and 601 bp DNA (a sequence with high nucleosome assembly potential). These differences including the absence of potentially regulatory C-terminal regions in the truncated construct and the varying length/structural properties of the DNA substrates introduce variables that substantially complicate direct comparative analysis of ATPase activity outcomes.

Separately, Douse et al. [4] demonstrated that the efficiency of HUSH complex-dependent epigenetic silencing decreases as MORC2’s ATP hydrolysis rate increases, implying an inverse relationship between ATPase activity and silencing function. Notably, our current work has not established a direct mechanistic link between MORC2 phase separation and its ATPase activity. Thus, we refrain from inferring that the effect of MORC2 phase separation on transcriptional repression is mediated through modulation of its ATPase function this remains an important question to address in future studies.

Finally, we have redesigned and expanded the experiments presented in Fig. 6 and Fig. S6 to directly link MORC2’s condensate-forming capacity with its transcriptional regulatory function.

**Reviewer #2 (Recommendations for the authors):**
Major concerns:(1) Unaddressed discrepancies with the previous study:(a) Inadequate discussion of Reference 22 and apparent contradictions. Notably, Reference 22 provides evidence for reduced ATPase activity upon DNA binding, in contrast to the current study's observations. Moreover, Reference 22 demonstrates that ATP hydrolysis (ATPase activity) is inversely associated with MORC2-mediated gene silencing, whereas this study concludes that 'the silencing function of MORC2 requires its ATPase activity'. These apparent contradictions warrant a more thorough discussion to reconcile the differences, including potential mechanistic explanations and experimental context that could account for the discrepancies. Additionally, the authors should discuss potential reasons why Ref. 22 may not have observed phase separation during MORC2 biophysical analysis. For instance, in Ref. 22, SEC-MALS was performed at 2 mg/mL (~16 µM) MORC2 FL in the presence of 150 mM NaCl, conditions that could influence phase behavior based on the current manuscript's results. Addressing whether differences in protein construct, buffer composition, or experimental design might account for this discrepancy would strengthen the discussion.

We thank the reviewer for pointing out the apparent discrepancies between our results and those reported in Ref. 22. We agree that these differences warrant explicit discussion, and we have revised the Discussion accordingly to clarify the experimental and conceptual distinctions between the two studies.

First, regarding the effect of DNA binding on ATPase activity, Ref. 22 examined MORC2 ATPase activity under conditions where MORC2 does not undergo detectable phase separation, whereas our ATPase assays were performed under conditions in which MORC2 readily forms condensates in the presence of DNA. We therefore propose that the observed increase in ATPase activity in our study may reflect a distinct biochemical regime in which phase separation and/or high local protein concentration modulates enzymatic activity. Importantly, our data do not exclude the possibility that DNA binding per se can inhibit ATPase activity under non-condensing conditions, as reported in Ref. 22.

Second, with respect to transcriptional repression, Ref. 22 reported an inverse correlation between ATP hydrolysis and MORC2-mediated silencing, whereas our study finds that ATPase activity is required for efficient repression. We suggest that these observations are not necessarily contradictory but may reflect different regulatory layers of MORC2 function. Specifically, ATP binding and hydrolysis may be required for MORC2 structural remodeling and chromatin engagement, while excessive or dysregulated ATP hydrolysis could impair stable silencing complexes, as suggested previously [4]. We now explicitly discuss this possibility in the revised manuscript.

Finally, we appreciate the reviewer’s suggestion regarding the absence of phase separation in Ref. 22. Indeed, SEC-MALS experiments in Ref. 22 were conducted at ~16 µM MORC2 in the presence of 150 mM NaCl (the purification condition is 500 mM NaCl, 10% glycerol), conditions that based on our phase diagrams—are close to or above the saturation concentration but also strongly influenced by ionic strength. This combination of factors explains why the UV peak from SEC-MALS is not indicative of a homogeneous sample [3].

(b) The DNA binding capacity of individual MORC2 domains was tested in Fig. 5. IDR appears to be the strongest DNA binder among others. Is this the effect of IDR being isolated from the rest of the protein? A recent paper (Tan, W., Park, J., Venugopal, H. et al. Nat Commun 2025) also investigated DNA binding capacity of different regions of MORC2 using hydrogen-deuterium exchange experiments and EMSA. Interestingly, it can be seen in Figure S9 that the DNA binding capacity of different regions changes when compared together to when in isolation (MORC2 1-603 vs 1-265; 1-495; 496-603). In line with the above, MORC2 IDR's interaction with DNA warrants additional investigation, taking the system as a whole to avoid misinterpretation arising from non-specific interactions.

We appreciate the reviewer’s insightful comments regarding domain-specific DNA binding and the potential caveats of studying isolated regions. In Figure 5, our EMSA analyses show that the isolated IDR exhibits the strongest DNA-binding signal among the tested fragments. We agree that this observation may, at least in part, reflect the removal of structural or regulatory constraints imposed by the full-length protein.

Consistent with the reviewer’s point, Tan et al. [5] demonstrated that DNA-binding behavior of MORC2 regions differs when analyzed in isolation versus in the context of larger constructs. We have now incorporated this comparison into the Discussion and explicitly note that DNA binding by the IDR should be interpreted as a contextual and potentially cooperative property rather than an autonomous function.

Importantly, our conclusions do not rely on the IDR acting as an independent DNA-binding module in vivo. Rather, we propose that the IDR contributes to DNA engagement and phase behavior within the architectural framework of full-length MORC2. We now emphasize this limitation and highlight the need for future studies that probe DNA binding in the context of intact MORC2 or minimally perturbed constructs.

(2) MORC2 DNA binding impacting phase separation and ATPase activity:While it is clear that MORC2: DNA interaction facilitates MORC2 phase separation, the impact on ATPase activity is not conclusive. First, they observe an opposite trend (compared to ref. 22) for DNA binding on MORC2's ATPase activity. Secondly, it is not clear if the increase in ATPase activity is mediated by DNA binding or phase separation. The ATPase activity was measured at 1 µM MORC2 protein concentration in the presence of DNA, where MORC2 appears to phase separate. To draw more definitive conclusions, additional controls are necessary. Specifically, a phase separation-deficient mutant (from this study) and a DNA-binding-deficient mutant (see ref. 22) should be included to disentangle the contributions of DNA binding and phase separation to ATPase activity. The choice of ATP-binding-deficient mutant N39A as a negative control seems inconclusive in this regard. Additionally, why is there an increase in ATP hydrolysis rate for the ATP-binding-deficient mutant in the presence of DNA, resulting in ATP hydrolysis rates similar to WT MORC2? This raises further questions about the underlying mechanism.

We agree with the reviewer that disentangling the contributions of DNA binding and phase separation to ATPase activity is challenging and that our current data do not fully resolve this issue. As noted, ATPase assays were performed at protein concentrations (1 µM) where MORC2 undergoes DNA-induced phase separation, making it difficult to distinguish whether enhanced ATP hydrolysis arises directly from DNA binding or indirectly from condensate formation.

We acknowledge that inclusion of additional mutants such as phase separation deficient or DNA-binding deficient variants would provide a more definitive mechanistic separation of these effects. However, generating and validating such mutants in a manner that preserves overall protein integrity is beyond the scope of the current study. Accordingly, we have revised the text to present our findings more cautiously and to frame the observed ATPase enhancement as a correlation rather than a causal mechanism.

Regarding the ATP-binding–deficient N39A mutant, we agree that its behavior in the presence of DNA raises interesting mechanistic questions. We now explicitly note this unexpected observation and discuss possible explanations, including partial ATP binding, altered oligomeric states, or indirect effects mediated by condensate formation.

(3) Dissecting the domain-specific contribution in MORC2 phase separation:(a) While in cellulo data indicate that the presence of IDR, NLS, CC3, and IBD is all essential for MORC2 condensate formation, it is not clear if this is the effect of the complex cellular environment or whether it is intrinsic for MORC2 phase separation ability. In lines 256-259, the authors suggest IDRa interaction with IBD may serve as a nucleation mechanism for LLPS. In other places, it has been mentioned that CC3 dimerization acts as a scaffold for condensate formation. It is not clear if all of these are essential for MORC2 phase separation, or one of them is essential while the other domain(s) facilitates the phase separation. Though Figure 3 provides a qualitative overview of the contribution of different regions in MORC2 phase separation in cellulo-influenced by the complex cellular environment and substrate interactions, the absolute domain contribution in phase separation would be better studied in vitro by quantitatively comparing phase diagrams (for example, c-sat vs temperature) of different domain deletion constructs.

We thank the reviewer for highlighting the distinction between intrinsic phase separation propensity and cellular context dependent effects. Our in cellular screening was designed to identify regions required for condensate formation under physiological conditions, where chromatin, binding partners, and macromolecular crowding are present. We agree that this approach does not directly quantify the intrinsic phase separation contribution of individual domains.

While CC3 dimerization, IDR–IBD interactions, and nuclear localization all contribute to condensate formation, our data do not imply that these elements are mechanistically equivalent. Rather, we propose that CC3 provides a structural scaffold, while IDR-mediated interactions lower the energetic barrier for condensation. We have revised the manuscript to clarify this hierarchical model and to avoid implying that all domains contribute equally or independently.

We agree that quantitative in vitro phase diagrams would provide valuable insight into intrinsic domain contributions. Whereas the MORC2ΔCC3-IBD (1–900) and CC3-IBD (900-1032) fragment fails to induce phase separation, the IDR mix CC3–IBD fragment drives robust phase separation; additionally, phase separation is entirely abrogated in the absence of domain–domain interactions. These observations collectively verify that phase separation is contingent on specific domain combinations and their interactions.

(b) Similarly, for line 228-231: 'Notably, condensates formed exclusively in the nucleus and not in the cytoplasm of transfected HeLa cells, suggesting that chromatin-associated nuclear factors, such as DNA, may contribute to the nucleation or stabilization of MORC2 condensates.' This is an important observation made by the authors. Since MORC2 readily phase separates in vitro under physiological conditions, it is important to discuss why MORC2 does not make condensates in the cytoplasm (in the case of MORC2deltaNLS). In this regard, how does the concentration of overexpressed EGFP-MORC2 constructs compare with in vitro tested droplets of MORC2?

We thank the reviewer for highlighting this important conceptual point. Although MORC2 readily undergoes phase separation in vitro under physiological buffer conditions, the absence of condensate formation in the cytoplasm of cells expressing MORC2ΔNLS underscores the importance of the nuclear environment in promoting MORC2 assembly.

The cytoplasm differs fundamentally from the nucleus not only in overall molecular composition but also in the availability of high-valency scaffolds such as chromatin. We propose that chromatin-associated components, particularly DNA, provide a platform that locally concentrates MORC2 and increases its effective valency, thereby facilitating nucleation or stabilization of condensates in the nucleus. In contrast, the cytoplasm lacks such scaffolds, even when MORC2 is expressed at appreciable levels. In cultured cells, MORC2 is seldom observed in the cytoplasm. While specific experimental contexts may facilitate its cytoplasmic localization, such observations are rarely reported [6]. In transfection-based systems, MORC2 predominantly displays droplet-like behavior in the nucleus. Notably, in endogenous EGFP–MORC2 chimeric mice, we detected punctate MORC2 structures in the neuronal cytoplasm of the brain and spinal cord. The functional significance and biophysical state of cytoplasmic MORC2 remain largely unexplored.

With respect to protein concentration, while EGFP-MORC2 is robustly expressed in cells, direct comparison between cellular expression levels and the protein concentrations used in vitro is inherently challenging. Importantly, in vitro phase separation is driven by bulk protein concentration under defined conditions, whereas in cells, effective local concentration and interaction valency are strongly shaped by spatial confinement and chromatin association. We have revised the manuscript text to emphasize this distinction and to avoid interpreting nuclear specificity as a purely concentration-dependent phenomenon.

(c) Lines 227-228: '... CW domain restricts condensate overgrowth or fusion', this inference is based on CTDdeltaCW puncta being larger in size (Figure 3a). However, in Figure 4h MORC2deltaIDRb and MORC2deltaIDRc also result in larger puncta. Making a final conclusion that the CW domain restricts condensate overgrowth or fusion warrants additional investigation.

We thank the reviewer for pointing out the limitation of our original conclusion. We agree that the enlarged puncta in both CTDΔCW (Figure 3a) indicate that condensate size regulation involves the CW domain was insufficiently rigorous.

Re-analysis of existing data identifies clear phenotypic disparities between the mutants: MORC2ΔIDRb/ΔIDRc mutants show two distinct phenotypes (reduced puncta number with enlarged size, or unchanged puncta number with uniform enlargement), and their total puncta area per cell is comparable to the WT. By contrast, CTDΔCW mutants display markedly larger puncta relative to the WT. Based on this distinction, we have revised our conclusion to a more cautious formulation: "These observations suggest that the CW domain may participate in regulating initial nucleation size and the exact molecular mechanisms require further investigation."

(4) MORC2 condensate-mediated gene silencing:This is one of the key investigations of this study where the authors evaluate the ability of MORC2 condensates to regulate gene silencing (transcriptional repression). The major concern here is that the authors are drawing their conclusion based on a CC3 domain deletion mutant of MORC2 and comparing it with wild-type MORC2. Notably, the CC3 domain is responsible for MORC2 dimerization, and as the authors quote, 'The dimeric assembly of CC3 is essential for maintaining the structural integrity of the protein', the absence of CC3 would have a direct impact on its function (such as ATPase activity). With these considerations, it is not clear whether the effect of CC3 domain deletion on gene regulation is an effect of no phase separation or a consequence of loss of function. This necessitates additional validation by including other controls, such as IBD domain deletion mutant, IDRa domain deletion mutant, where the phase separation is impeded without affecting dimerization.

We appreciate the reviewer’s concern regarding the interpretation of CC3 deletion experiments. We agree that CC3 deletion affects both dimerization and phase separation, complicating attribution of gene regulatory effects solely to condensate formation. Our intention was not to claim that loss of repression arises exclusively from impaired phase separation, but rather to demonstrate that disrupting condensate-dynamic capacity correlates with impaired silencing.

To directly address these concerns, we have performed a series of new experiments specifically designed to decouple condensate formation, condensate dynamics, and protein abundance, thereby allowing us to more rigorously interrogate the functional relevance of MORC2 condensates.

First, to overcome the limitation of domain deletions which may affect MORC2 function beyond phase separation we introduced a micropeptide-based kill switch (KS) to the C terminus of MORC2. This strategy has recently emerged as a powerful approach to selectively reduce condensate dynamics without disrupting protein expression, folding, or domain architecture [1]. Importantly, unlike CC3 or IDRa deletions, MORC2+KS robustly form nuclear condensates but exhibits markedly reduced internal dynamics, as demonstrated by FRAP analyses showing minimal fluorescence recovery after photo bleaching (Fig. 6a-c). This strategy therefore allows us to perturb condensate material properties independently of MORC2 domain integrity.

Second, we systematically compared the transcriptional consequences of rescuing MORC2-knockout HeLa cells with MORC2FL, condensation-deficient mutants (ΔCC3 and ΔIDRa), and the dynamics-defective MORC2+KS (Fig. 6d). Despite being expressed at substantially higher levels than MORC2FL (Fig. 6e), all three mutants showed a striking and consistent failure to restore MORC2-dependent transcriptional regulation (Fig. 6f-h). This effect was particularly pronounced for transcriptionally repressed genes, including two sets of high-confidence MORC2 targets reported in prior studies (Fig. 6i and Fig. S10). These findings demonstrate that neither increased protein abundance nor the mere presence of condensate-like structures alone is sufficient to restore MORC2 function.

Third, our data instead support a model in which both soluble MORC2 complexes and dynamic MORC2 condensates are required for full transcriptional activity. While soluble MORC2 is likely involved in target recognition and complex assembly, our results indicate that proper condensate formation and critically, condensate dynamics are essential for effective transcriptional repression and activation. The inability of the MORC2+KS mutant to rescue transcriptional defects, despite intact condensate formation, points away from a model in which MORC2 condensates represent only microscopically visible byproducts of MORC2 activity.

We believe these new data strengthen the manuscript by pairing the detailed mechanistic dissection of MORC2 phase separation with direct functional evidence, enhancing the conceptual impact and biological significance of the study.

(5) Uncertain impact of pathogenic MORC2 mutations:Line 356-365: While the statements such as "disease-associated mutations primarily affect enzymatic and phase behaviors rather than DNA affinity" and "these findings provide mechanistic insight into how specific mutations may contribute to distinct pathological outcomes" are conceptually compelling, the data presented in Figure 7b-d do not appear to fully support these conclusions. For many of the mutants, the differences from WT across key parameters-condensation, ATPase activity, and DNA binding-are either modest or statistically insignificant. As such, drawing a unified mechanistic conclusion from these datasets may overstate what the data actually support.

We agree that the effects of disease-associated MORC2 mutations described in Fig. 7 are modest and, in some cases, statistically insignificant. Our intention was to document observable trends rather than to propose a unified mechanistic framework. We have revised the manuscript to temper these conclusions and to emphasize the descriptive nature of these data.

(6) Important conceptual clarifications:(a) Intrinsically disordered regions (IDRs) are not synonymous with phase separation. As the authors show, it is a combination of IDR-mediated interactions and CC3 dimerization that contributes towards the phase separation of MORC2. While IDRs can act as scaffolds for multivalent weak interactions that may promote biomolecular condensate formation, many IDRs serve other roles-such as mediating transient interactions, signaling, or regulatory functions-without undergoing phase separation. Researchers should avoid generalizing the assumption that the mere presence of IDRs in a protein implies its ability for phase separation. In this regard, authors should consider restructuring some of their generalized statements: Line 87-88: 'Recent studies suggest that intrinsically disordered regions (IDRs) can drive liquid-liquid phase separation (LLPS)' and Line 159-161: 'we noticed a long unstructured region at its C-terminus (Fig. S1b), a characteristic often associated with proteins capable of phase separation'.

We agree that IDRs are not synonymous with phase separation and have revised the Introduction to avoid generalized statements. The revised text now emphasizes that IDRs can contribute to phase separation in a context-dependent manner and act in concert with structured oligomerization domains such as CC3-IBD.

(b) Liquid-liquid phase separation: I would suggest switching the phrase to just phase separation. The rationale is that the in vitro studies of MORC2 (FRAP, droplet imaging) do not show liquid-like behavior, but perhaps liquid-solid. The FRAP studies suggest liquid-like behavior for some of the constructs. Given the differences in viscoelastic properties across the in vitro and in cellulo studies, it is better to generalize to "phase separation". Movies for droplet fusion and FRAP, wherever applicable, would be much appreciated. As the nature of in vitro MORC2 droplets appears different than in cells, movie representations of the above would enable readers to better assess the viscoelastic nature of the droplets (whether liquid, gel, etc).

We appreciate the reviewer’s insight regarding the viscoelastic properties of MORC2. Our experimental data indeed show a disparity in dynamics between the two environments: while in vitro MORC2-FL condensates exhibit relatively low internal mobility, the in cellulo MORC2-FL puncta display high dynamics, characterized by rapid internal recovery in FRAP assays and droplet fusion events (Fig. S2f).

This contrast suggests that the intracellular microenvironment plays a critical role in regulating the material state of MORC2 condensates. Consequently, we have focused on providing in vivo fusion data, as we believe in vitro characterizations (such as fusion or FRAP under various artificial conditions) may not faithfully represent the physiological behavior of MORC2. We have revised the manuscript to use the more general term “phase separation” or “condensation” and have added a discussion on these limitations to avoid overinterpreting the material properties observed in vitro.

(7) Methods:(a) Figure 6 S2b: If phase separation occurs at, say, 1.8 µM protein concentration, this indicates that the protein has reached its saturation concentration (c-sat). Beyond c-sat, any additional protein should partition into the dense phase, while the concentration of the dilute phase remains constant. However, in this figure, the dilute phase concentration appears to increase with increasing total protein concentration, which is inconsistent with expected phase separation behavior. As the methods section does not have any sub-section for the sedimentation assay, it becomes difficult to understand how this experiment was performed, whether there is any technical discrepancy in the way soluble and pellet fractions were handled and processed for loading onto the gels. This is also the case with Figure 3d.

We thank the reviewer for carefully examining the sedimentation assay and for raising this important conceptual point. We agree that, for an ideal two-phase system at thermodynamic equilibrium, the concentration of the dilute phase is expected to remain constant once the saturation concentration (c-sat) is reached.

In our study, the sedimentation assay was used as an operational readout to assess concentration-dependent partitioning rather than to quantitatively define equilibrium phase boundaries. The assay involves centrifugation-based separation of supernatant and pellet fractions followed by SDS–PAGE analysis, and therefore does not necessarily report the equilibrium concentrations of coexisting dilute and dense phases. In particular, this approach can be influenced by incomplete physical separation of phases, kinetic trapping, and redistribution of material during handling, especially in systems where condensate maturation or internal reorganization occurs on longer timescales.

Consequently, the apparent increase in the supernatant fraction with increasing total protein concentration likely stems from kinetic limitations and inherent technical constraints of the sedimentation assay, rather than a genuine deviation from classical phase separation behavior. These caveats are now explicitly clarified in the Methods section, with similar limitations of centrifugation-based assays for defining equilibrium phase behavior of biomolecular condensates reported previously.

(b) Figure 4: The NMR comparisons appear to be primarily qualitative, lacking quantitative analyses such as chemical shift perturbation (CSP) and intensity ratio plots, which would offer deeper mechanistic insights. The NMR spectra detailing interactions among the IDR domains need to be quantified.

We thank the reviewer for the suggestion. We have now performed quantitative CSP analyses for the NMR data shown in Fig. 4, and the corresponding CSP plots have been added to the revised manuscript (Fig. S7).

As expected for interactions mediated by intrinsically disordered regions involved in phase separation, the observed CSPs are generally small. Notably, the CSP profile of IDRa closely matches that observed for the full-length IDR, whereas IDRb and IDRc show minimal perturbations. These results indicate that the interaction is primarily mediated by IDRa, with little contribution from the remaining regions.

Peak intensity analyses were also examined but did not reveal additional residue-specific trends. Together, the quantitative CSP data support our conclusion that the interaction is weak, dynamic, and region-specific, consistent with an IDR-driven, phase-separation-related mechanism. We add this statement in method: CSPs were calculated in Hz at 600 MHz using the following equation:\begin{document}$$\displaystyle \Delta v=\left(\left(\Delta^{1} \mathrm{H} \times 600\right)^{2}+\left(\Delta^{13} \mathrm{C} \times 150\right)^{2}\right)^{1 / 2}$$\end{document}

Minor comments:(1) Line 59-60: The Authors mention the HUSH-complex and then the MORC protein family, but do not discuss the relation between the two.

We thank the reviewer for this comment. We have revised the Introduction to explicitly state that MORC2 may serve as a component of the HUSH complex and to clarify the functional relationship between MORC family proteins and HUSH-mediated transcriptional repression.

(2) Line 74: 'Despite their structural similarities...', similarities between what all?

We agree that this statement was ambiguous. We have revised the text to explicitly specify that the comparison refers to structural similarities among MORC family members.

(3) Line 75: 'MORC-mediated repression remains...', this is the first time the word 'repression' is mentioned in the text and directly as an outstanding question.

We have revised the Introduction to introduce the concept of transcriptional repression earlier and to provide appropriate context before posing it as an outstanding question.

(4) The third paragraph does address issues in comments 1 and 3 to some extent, but the introduction needs some restructuring to provide a proper flow of information.

We agree that the Introduction required restructuring. We have revised this section to improve logical flow, better integrate prior studies, and more clearly articulate the motivation and scope of the present work.

(5) Line 83-85: How does the presence of IDRs suggest potential regulatory mechanisms?

We have revised this sentence to clarify that IDRs may contribute to regulatory mechanisms by enabling multivalent and dynamic interactions, rather than implying that IDRs inherently confer regulatory function or phase separation capability.

(6) Line 106-107: 'To determine whether MORC2 has N- and C-terminal dimerization interfaces similar to those...', reference 14 has already established that CC3 (denoted as CC4 in ref 14) is responsible for dimerization. Consider acknowledging their work in this regard?

We thank the reviewer for this reminder. We have now explicitly acknowledged Ref. 14, which previously established the role of CC3 (denoted CC4 in that study) in MORC2 dimerization.

(7) Lines 117-122: Are the authors comparing morphology from negative stain EM with AlphaFold predicted structure (Figure S1a and S1b)? If so, providing a zoomed-in inset from Figure S1a would be helpful.

Yes, the comparison was intended to relate the negative-stain EM morphology to the AlphaFold-predicted architecture. We have added a zoomed-in inset in Fig. S1a to facilitate clearer comparison.

(8) Line 152-153: '...even under varying physiological conditions', what are these varying conditions? Are the authors trying to point towards any of their specific results?

We have revised this phrase to explicitly refer to variations in salt concentration and protein concentration tested in our in vitro assays.

(9) Line 154-155: 'The dimeric assembly of CC3 is essential for maintaining the structural integrity of the protein', if it has been established, then please provide a reference.

We thank the reviewer for this suggestion. For MORC family proteins, C-terminal coiled-coil–mediated dimerization is necessary for correct homodimer formation and functional stability (Xie et al., 2019, Cell Commun Signal. 17:160, Ref 14 in the revised manuscript).

(10) Line 159-161: 'we noticed a long unstructured region at its C-terminus (Figure S1b), a characteristic often associated with proteins capable of phase separation25.', again authors are generalizing a statement which is, in most cases, context-dependent. For example, ref 25 mentions that unstructured regions or IDRs serve as a scaffold for multivalent interactions.

We agree with the reviewer and have revised this sentence to avoid generalization. The revised text now emphasizes that IDRs may facilitate multivalent interactions in a context-dependent manner, rather than being intrinsically indicative of phase separation. Additionally, we have explicitly cited the mechanistic insight from Reference 25 that IDRs serve as scaffolds for multivalent interactions, to strengthen the logical link between the structural feature and its potential functional relevance.

(11) Methods section for NMR (Line 665-667) mentions that nucleotides were added to a final concentration of 10 mM. There is no figure or section for MORC2 NMR with added nucleotides/DNA.

We thank the reviewer for pointing this out. The nucleotide (ATP) addition was part of preliminary NMR trials and is not directly associated with the figures presented. We have deleted this in the Methods section to avoid confusion.

(12) Line 285-294: Authors compare the effect of DNA binding on the phase separation of both MORC2FL and MORC2 CTDdeltaCW and conclude that DNA-induced condensation is primarily mediated through interactions with the IDR-NLS region. This appears not to be backed by proper control experiments. The authors do not show whether DNA binding mediates any phase separation for the isolated NTD or not? Similarly, what is the effect of DNA binding on MORC2 deltaIDR?

We thank the reviewer for this insightful comment and agree that additional controls are essential for rigorously dissecting the contribution of DNA binding to MORC2 phase separation. Our interpretation that DNA-enhanced condensation is primarily mediated through the IDR–NLS region was based on comparative analyses of MORC2FL and MORC2 CTDΔCW, together with EMSA results demonstrating that DNA binding activity is conferred by the IDR–NLS–containing region. We acknowledge, however, that DNA binding alone is not sufficient to infer phase separation behavior.

To address this point, we have performed additional analyses using the isolated NTD’ (residues 1–536) and MORC2 ΔIDR–NLS mutants (Fig. S6). The isolated NTD’ exhibited detectable DNA binding [4] but did not undergo DNA-induced condensation under conditions while MORC2FL or MORC2 CTDΔCW (residues 537-1032) readily formed condensates, indicating that DNA binding by itself is insufficient to drive phase separation. In parallel, MORC2 ΔIDR–NLS mutants showed severely compromised solubility and stability in vitro, which limited their quantitative characterization in phase separation assays. Nevertheless, under the conditions tested, these mutants did not display DNA-enhanced condensation comparable to MORC2FL.

Taken together, these observations support a model in which the IDR–NLS region plays a critical role in coupling DNA binding to condensation, while additional domains are required to sustain robust phase separation. We have revised the manuscript text to clarify the experimental scope and to avoid overinterpreting the contribution of DNA binding in the absence of fully reconstituted control systems.

(13) How did the authors assign the backbone amide NMR chemical shifts for MORC2?

Backbone assignments of MORC2 IBD (1004-1032) were obtained using SOFAST versions of standard triple-resonance experiments, including HNCACB and CBCACONH, recorded at 298 K. Residual assignment ambiguities were resolved using [15] N-edited HMQC-NOESY-HMQC spectra.

(14) Line 256: 'The partial compaction of IDRa...', what does the author mean here with 'partial compaction'? How did they measure compaction here?

Regarding the term “partial compaction” mentioned previously, we apologize for the typographical error this phrase was erroneously used in place of “key component”.

(15) Line 312-315: Why is there even a MORC2 readout for MORC2 KO cells with only EGFP? Also, the authors suggest that IDR deletion may impair mRNA stability or transcription; however, the expression levels of MORC2 deltaIDR and MORC2 deltaCC3 do not appear drastically different in Figure 3a.

We thank the reviewer for raising these points. The apparent MORC2 signal in MORC2 knockout cells transfected with EGFP alone is due to the presence of residual MORC2 mRNA. Although CRISPR–Cas9–mediated knockout introduces a frameshift that prevents MORC2 protein expression, the mRNA can still be detected by RNA-seq. This is because nonsense-mediated decay (NMD), which targets transcripts with premature stop codons for degradation, is not always 100% efficient. Therefore, some MORC2 transcripts remain and produce detectable RNA-seq reads, even though no functional protein is expressed.

Regarding the apparent discrepancy in expression levels, Fig. 3a displays only EGFP-positive cells, within which the fluorescence intensity of MORC2ΔIDR and MORC2ΔCC3 appears comparable to that of WT MORC2. However, the overall fraction of EGFP-positive cells is markedly reduced for these mutants compared to WT. Thus, while expression levels among successfully transfected cells are similar, fewer cells express detectable levels of the ΔIDR or ΔCC3 constructs across the total population. We therefore interpret this reduction in EGFP-positive cell fraction as reflecting impaired expression efficiency of these mutants, potentially arising from altered transcriptional output, mRNA stability, or protein stability. We have revised the manuscript text to clarify this distinction and to avoid overinterpreting the underlying mechanism in the absence of direct measurements.

**Author response image 1. sa4fig1:** 

EGFP, EGFP–MORC2 (FL), EGFP–MORC2 (ΔCC3), and EGFP–MORC2 (ΔIDR) were re-expressed in MORC2-knockout HeLa cells. Confocal imaging revealed that full-length MORC2 formed condensates in the nucleus, whereas mutants lacking either the CC3 or IDR domain failed to exhibit such behavior. Notably, under identical experimental conditions, we observed a marked reduction in the transfection efficiency of the EGFP-MORC2 (ΔIDR) construct. In contrast to the other variants, EGFP signals for ΔIDR were detectable in only a small fraction of the total cell population, despite consistent DNA loading and protocol synchronization. This observation suggests that the IDR might be required not only for biomolecular condensation but also for maintaining the steady-state levels of the MORC2 mRNA/protein or overall cellular fitness.

(16) Line 330: 'MORC2 deltaCC3 failed to repress any of the 18 downregulated targets...'. This does not appear to be entirely true as repression of some targets (LBH, TGFB2, GADD45A) are closer to MORC2 FL than the EGFP control.

We thank the reviewer for pointing out this inconsistency and for highlighting the need for precise wording. We have updated the dataset and revised the text to describe the results more accurately. We now describe that the mutants impair MORC2FL-mediated transcriptional regulation, consistent with the overall trend observed across these target genes.

(17) Line 347-350: Based on the percent of cells with condensates, the authors conclude that CMT2Z-linked E236G and SMA-linked T424R mutants promote MORC2 phase separation. Again, the effect of these mutations on MORC2 condensation in cells may be direct or indirect. This can be investigated by comparing the in vitro effect of these mutations on MORC2 phase separation.

We thank the reviewer for raising this important point and fully agree that the effects of disease-associated MORC2 mutations on condensate formation in cells may arise from either direct alteration in intrinsic phase separation propensity or indirect influences mediated by the cellular environment.

In our study, disease-associated MORC2 mutants were assessed for condensate formation in HEK293F cells. Attempts were made to characterize these mutants in vitro; however, the E236G mutant exhibited markedly reduced solubility and stability upon purification, which precluded reliable in vitro phase separation analysis. We therefore evaluated the impact of E236G in cells and found that this mutation significantly impaired the dynamics of nuclear MORC2 condensates. For the T424R mutant, we note that its intracellular condensates displayed FRAP recovery kinetics comparable to those of WT MORC2, suggesting broadly similar dynamic properties of the assemblies formed in cells, but not necessarily implying a direct enhancement of intrinsic phase separation.

In light of these considerations, we have revised the text in Lines 347–350 to avoid attributing a direct causal role of these mutations in promoting MORC2 phase separation. Instead, we now describe the observed increase in the fraction of cells containing condensates as a descriptive cellular correlation. We further emphasize that systematic in vitro characterization of disease-associated MORC2 mutants will be required to distinguish direct from indirect effects and represents an important direction for future investigation.

(18) The discussion section lacks referencing to individual figures in the results section as well as previous literature.

We agree with the reviewer that the Discussion would benefit from clearer integration with both the Results figures and prior literature. In the revised manuscript, we have substantially restructured the Discussion to explicitly reference key figures when interpreting experimental findings and to more clearly distinguish conclusions drawn from specific datasets. In addition, we have expanded citations to previous studies where relevant, particularly in the context of MORC2 DNA binding, ATPase regulation, chromatin association, and disease-linked mutations. These revisions aim to better situate our findings within the existing literature and to guide readers more clearly between experimental observations and their interpretation.

**Reviewer #3 (Public review):**
Summary:The manuscript by Zhang et al. demonstrates that MORC2 undergoes liquid-liquid phase separation (LLPS) to form nuclear condensates critical for transcriptional repression. Using a combination of in vitro LLPS assays, cellular studies, NMR spectroscopy, and crystallography, the authors show that a dimeric scaffold formed by CC3 drives phase separation, while multivalent interactions between an intrinsically disordered region (IDR) and a newly defined IDR-binding domain (IBD) further promote condensate formation. Notably, LLPS enhances MORC2 ATPase activity in a DNA-dependent manner and contributes to transcriptional regulation, establishing a functional link between phase separation, DNA binding, and transcriptional control. Overall, the manuscript is well-organized and logically structured, offering mechanistic insights into MORC2 function, and most conclusions are supported by the presented data. Nevertheless, some of the claims are not sufficiently supported by the current data and would benefit from additional evidence to strengthen the conclusions.

Thank you for your insightful review and constructive suggestions, which have been invaluable in refining our manuscript.

The following suggestions may help strengthen the manuscript:Major comments:(1) The central model proposes that multivalent interactions between the IDR and IBD promote MORC2 LLPS. However, the characterization of these interactions is currently limited. It is recommended that the authors perform more systematic analyses to investigate the contribution of these interactions to LLPS, for example, by in vitro assays assessing how the IDR or IBD individually influence MORC2 phase separation.

We appreciate the reviewer’s insightful comment regarding the characterization of IDR–IBD interactions. In this study, we combined NMR spectroscopy, domain deletion analysis (in vivo), and in vitro phase separation assays to demonstrate that interactions between the IDR and IBD contribute to MORC2 condensate formation. To systematically assess the individual contributions of the IDR and IBD to MORC2 phase separation, we performed in vitro reconstitution assays using purified domain constructs (Fig. S6). Neither the isolated IDR nor the IBD alone exhibited phase separation under buffer conditions approximating the physiological environment, indicating that each domain is individually insufficient to drive condensation. Upon the addition of 10% PEG8000, phase separation was selectively observed for the IDR but not for the IBD, suggesting that the IDR possesses an intrinsic propensity for phase separation that can be enhanced by crowding molecular. Importantly, when the IDR and IBD were mixed, phase separation was robustly induced, supporting a model in which cooperative inter-domain interactions between the IDR and IBD promote MORC2 condensation. In the absence of PEG, no phase separation was observed for the IDR–IBD mixture. These observations imply that IDR–IBD interactions cannot drive phase separation on their own, but require cooperation with CC3-mediated dimerization to achieve this process, which is the central point we wish to emphasize.

(2) The authors mention that DNA binding can promote MORC2 LLPS. It is recommended that they generate a phase diagram to systematically assess how DNA influences phase separation.

We agree that constructing a full phase diagram would provide a more systematic evaluation of the effect of DNA on MORC2 phase separation. In the current study, we assessed DNA-dependent condensation across multiple protein and DNA concentrations, which consistently showed that DNA enhances MORC2 phase separation. At low protein concentration (0.5 µM), phase separation requires sufficient DNA, whereas increasing either DNA or protein concentration promotes liquid droplet formation. At high DNA and protein concentrations, amorphous structures dominate, indicating a transition away from dynamic assemblies. We have clarified this point in the Results and Discussion sections and now note that a comprehensive phase diagram analysis represents an important direction for future work.

(3) The authors use the N39A mutant as a negative control to study the effect of DNA binding on ATP hydrolysis. Given that N39A is defective in DNA binding, it could also be employed to directly test whether DNA binding influences MORC2 phase separation.

We thank you for your constructive suggestions. The purified wild-type MORC2(1–603) exhibited weak but detectable ATPase activity, whereas the N39A mutant was completely inactive [5]. Based on this characteristic, the N39A mutant was used as a negative control for the ATP-binding-deficient mutant in this study [3]. However, no evidence has been provided to demonstrate that the N39A mutant is defective in DNA binding. Importantly, both our results and previous studies [5-6] indicate that MORC2 engages DNA via multiple domains, suggesting that a single-point mutation is unlikely to significantly compromise its overall DNA-binding capacity.

(4) Many of the cellular and in vitro LLPS experiments employ EGFP fusions. The authors should evaluate whether the EGFP tag influences MORC2 phase separation behavior.

We appreciate the reviewer’s concern regarding the potential influence of the EGFP tag. The use of EGFP fusions in our study was primarily to maintain consistency with the in-cell experiments. Importantly, we confirmed that EGFP alone does not undergo phase separation in cells, and this observation is consistent with previous studies [7]. Additionally, in vitro phase separation of MORC2 was independently validated using Cy3–labeled CTD (Fig. S5), which recapitulated the condensate formation seen with EGFP-fused protein. Together, these results indicate that the EGFP tag does not significantly influence MORC2 phase separation, supporting the validity of our conclusions.

**Reviewer #3 (Recommendations for the authors):**
(1) The authors claim to have obtained nucleic acid-free protein, but no data are provided to support this assertion. It is recommended that they include appropriate validation to confirm the absence of nucleic acids.

We thank the reviewer for highlighting this point. To validate that the purified MORC2 protein is indeed free of nucleic acid contamination, we have additional experimental evidence (e.g., A260/280 measurements, agarose gel analysis, or EMSA in Fig. 5), which has been added to the Methods section and Table S2.

Note: Agarose gel analysis for MORC2 constructs to confirm the absence of nucleic acids. The pET32 vector as the positive control, the protein preparation for analysis is 0.05 mg. E means *E. coli* and H means HEK293F.

(2) The FRAP recovery curves are not normalized to 0, making comparison difficult. The authors should normalize the post-bleach intensity to 0 and re-plot the curves to allow a more standard interpretation of mobile fractions.

We agree with the reviewer and have now normalized the FRAP recovery curves by setting the post-bleach intensity to 0. The revised plots are presented in the Figures (2f, j, l; 6c, 7f), allowing for more direct comparison of mobile fractions across different conditions.

(3) The HSQC spectra for IBD appear inconsistent: the peak positions in Fig. 4C do not align with those shown in panels D-F. The authors should verify the spectral assignments and ensure consistency across figures.

We thank the reviewer for pointing this out. The apparent inconsistency arose from the fact that different spectral regions were displayed in Fig. 4c versus Fig. 4d-f for visualization purposes, which may have given the impression of mismatched peak positions. The spectral assignments themselves are consistent across all panels.

To avoid confusion, we have now adjusted the spectral window shown in Fig. 4c to match that used in Fig. 4d-f. The revised figure ensures consistent presentation of the same spectral region across all panels.

Reference:

(1) Zhang, Y., Stöppelkamp, I., Fernandez-Pernas, P. et al. Probing condensate microenvironments with a micropeptide killswitch. Nature 643, 1107–1116 (2025).

(2) Fendler NL, Ly J, Welp L, et al. Identification and characterization of a human MORC2 DNA binding region that is required for gene silencing. Nucleic Acids Res.53(4):gkae1273 (2025).

(3) Tchasovnikarova, I., Timms, R., Douse, C. et al. Hyperactivation of HUSH complex function by Charcot–Marie–Tooth disease mutation in MORC2. Nat Genet 49, 1035–1044 (2017).

(4) Douse, C. H. et al. Neuropathic MORC2 mutations perturb GHKL ATPase dimerization dynamics and epigenetic silencing by multiple structural mechanisms. Nat Commun 9, 651 (2018).

(5) Tan, W., Park, J., Venugopal, H. et al. MORC2 is a phosphorylation-dependent DNA compaction machine. Nat Commun 16, 5606 (2025).

(6) Sánchez-Solana B, Li DQ, Kumar R. Cytosolic functions of MORC2 in lipogenesis and adipogenesis. Biochim Biophys Acta. 1843(2):316-326 (2014).

(7) Li, C.H., Coffey, E.L., Dall’Agnese, A. et al. MeCP2 links heterochromatin condensates and neurodevelopmental disease. Nature 586, 440–444 (2020).